



# Diagnosis of winter precipitation types using Spectral Bin Model (SBM): Comparison of five methods using ICE-POP 2018 field experiment data

Wonbae Bang[1], Jacob T. Carlin[2], Kwonil Kim[4], Alexander V. Ryzhkov[2], Guosheng Liu[3], and GyuWon Lee[1]

[1]BK21 Weather Extremes Education & Research Team, Department of Atmospheric Sciences, Center for Atmospheric REmote sensing (CARE), Kyungpook National University, Daegu, Republic of Korea

[2]Cooperative Institute for Severe and High-Impact Weather Research and Operations, The University of Oklahoma, and NOAA/OAR National Severe Storms Laboratory, Norman, Oklahoma, USA

[3]Department of Earth, Ocean and Atmospheric Science, Florida State University, Tallahassee, Florida, USA

[4]School of Marine and Atmospheric Sciences, Stony Brook University, New York, USA

*Correspondence to*: GyuWon Lee (gyuwon@knu.ac.kr)





**Abstract**

Winter precipitation types (WPTs) are controlled by many factors, including thermodynamic and microphysical processes. Therefore, realistically simulating interactions between precipitation particles and the atmosphere is important when diagnosing the WPT. In the present study, we analyze the performance of the one-dimensional spectral bin model (SBM) developed by Carlin and Ryzhkov (2019), which simulates the change in the physical characteristics of precipitation particles

of various sizes as they fall from the cloud top to the ground and diagnoses surface WPT. We compare the performance of the SBM and four other diagnostic methods that use the following variables: 1) atmospheric thickness, 2) wet-bulb temperature, 3) temperature and relative humidity, and 4) wet-bulb temperature and low-level lapse rate. Three reference WPTs (snow [SN], rain [RA], and RASN) are obtained from particle size velocity (PARSIVEL) disdrometer data using a newly proposed decision algorithm. The results show that the SBM has the highest overall skill score for winter precipitation, especially at the mountain

sites. In contrast, the skill score of the SBM is lower than the other methods for RA. These results indicate that the SBM simulations tend to underestimate melting compared to observations. We thus explore the effects of the SBM's microphysics scheme on the extent of melting in cases of misdiagnosed RA. An optimized SBM that uses the climatological snow density-diameter relationship for the Pyeongchang region produces an increased amount of melting and achieves an improved skill score compared to the original SBM, which uses climatological relationships for Colorado region.

**1. Introduction**

There is a complex variety of winter precipitation types (WPTs) such as rain (RA), snow (SN), rain and snow (RASN), ice pellets (IPs), freezing rain (FZRA), and a mixture of ice pellets and freezing rain (IPFZRA). Various thermodynamical and microphysical processes can determine surface WPTs in nature. Some microphysical processes, such as melting, freezing, vaporization, and sublimation, change the phase and mass of precipitation particles and are diabatic thermodynamic processes.

Other microphysical processes, such as riming and aggregation, modify particle size distributions (PSDs), habits, and the physical characteristics of individual particles such as their fall velocity and density (Heymsfield, 1972; Pruppacher and Klett, 1997; Libbrecht, 2001; Barthazy and Schefold, 2006; Lee et al., 2015; Gong et al., 2020; Vázquez-Martín et al., 2020). Aggregation processes widen PSDs by increasing the size of particles, while riming processes increase the terminal fall velocity and density of the particles. Thus, the complexity of these processes should be accounted for when seeking to accurately

diagnose WPTs.

Several simple empirical methods are commonly used to predict WPTs based on empirical relationships between specific meteorological variables and WPTs. For example, the atmospheric thickness can be used to classify WPTs. Because this thickness is proportional to the mean virtual temperature ($T_v$) between two layers, a higher thickness is associated with a higher possibility of melting. Different thresholds for atmospheric thickness are used depending on the region under investigation

(Koolwine, 1975; Stewart and King, 1987; Bluestein, 1993; Lee et al., 2014). In addition, nomograms of relative humidity (RH) and temperature ($T$) on the ground can be used to determine the WPT. Matsuo et al. (1981) proposed RH-$T$ relationships



to distinguish three WPTs (RA, RASN, and SN), and Lee et al. (2014) subsequently modified this using observational data. The wet-bulb temperature ($T_w$) can also be used as a predictor. $T_w$ is defined as the temperature of the air when brought to saturation by the evaporation of water. $T_w$ is a better-conserved quantity than $T$, which makes it useful for short-range

predictions. Häggmark et al. (2000) developed a probability density function (PDF) for SN as a function of the surface $T_w$. Recently, joint probability distributions for SN using $T_w$ and the low-level lapse rate (i.e., the rate of change in the temperature from the surface to 500 m above ground level [AGL]; °C km$^{-1}$) have been proposed based on an analysis of global statistical data (Sims and Liu, 2015). By including the low-level lapse rate, the scheme proposed by Sims and Liu (2015) takes into account situations where the melting of ice particles begins while they are falling, which is especially important for conditions

that include low-level temperature inversions. However, because this scheme was developed using global data without regional and/or synoptic weather dependence, it is only valid when used in a globally averaged manner. The validity for the regions of this study has not been investigated in Sims and Liu (2015). In addition to those described here, many other WPT diagnostic methods based on the environment have been proposed (e.g., Ramer, 1993; Baldwin et al., 1994; Bourgouin, 2000; Schuur et al., 2012).

Other studies have attempted to predict WPTs using the environmental data combined with an explicit microphysical model (e.g., Reeves et al., 2016). This approach is motivated by the fact that the rate of change between phases varies with the particle size; for example, small particles may entirely melt while larger particles remain predominantly ice. This subsequently affects refreezing because the threshold for $T$ needed to initiate refreezing depends on whether an ice nucleus remains in the particle or whether it is entirely liquid. As such, the accurate diagnosis of WPTs at the surface requires consideration of these processes

as a function of particle size, particularly for a mixture of WPTs (e.g., RASN and IPFZRA).

The one-dimensional spectral bin model (SBM) proposed by Reeves et al. (2016) separates the precipitation PSD into various bins and calculates the phase change for each of these bins at sequential height intervals using heat balance equations that depend on the environmental $T$ and humidity (Rogers and Yau, 1989; Pruppacher and Klett, 1997). The resultant WPT (RA, SN, RASN, IP, FZRA, or IPFZRA) is predicted based on the relative fractions of ice and liquid at the surface (see Sect.

3.2 for more details). The original formulation (Reeves et al. 2016) used a fixed PSD of aggregated SN particles with various degrees of riming and was mass-conserving by only considering melting and refreezing. Carlin and Ryzhkov (2019) expanded the microphysical component of the SBM to include varying PSDs, multiple particle habits, and sublimation and evaporation. The addition of sublimation and evaporation is predicated on the idea that these processes may effectively eliminate the hydrometeor mass at the low end of the PSD, thus affecting the resulting classification. Evaluation of the SBM optimized for

the United States (Reeves et al., 2016) revealed that the model was highly skilled in discriminating FZRA and IPs, but achieved slightly lower scores for SN and RA when compared to other algorithms that rely only on environmental metrics (Ramer, 1993; Baldwin et al., 1994; Bourgouin, 2000; Schuur et al., 2012).

Intensive observation networks employing a variety of instruments (e.g., disdrometers, weighing gauges, radar, lidar, and rawinsondes) were established at many sites along the South Korean coastline and across the Taebaek mountains during the

International Collaborative Experiments for Pyeongchang 2018 Olympic and Paralympic Winter Games campaign (ICE-POP





2018, Lee and Kim, 2019; Gehring et al., 2020). The Taebaek mountain ranges are complex, experiencing sudden changes in the surface conditions due to the effects of the relatively warm ocean and cold mountainous terrain frequently occurring in this region. Thus, the winter precipitation in this region is affected by different synoptic patterns, orographic effects, and air-sea interactions (Nam et al., 2014; Kim et al., 2019). There are also many local and small-scale phenomena to consider, such as

the occurrence of cold pools due to the development of coastal fronts, the formation of inversion layers (ILs) aloft as a result of these cool pools, and greater low-level thermal instability due to warm and moist advection from the ocean. Nevertheless, although the accurate diagnosis of the WPT is challenging in this region, the intensive observation data density available due to the ICE-POP network allows for an extensive evaluation and optimization of previously proposed WPT diagnosis methods.

In this study, we aim to compare the performance of the SBM (Reeves et al., 2016; Carlin et al., 2019) with empirical

methods in terms of dianosing the WPT using observations from rawinsondes. The four empirical approaches tested are the 1000-850 hPa thickness ($H_{850}$), RH-$T$ nomogram (Lee et al., 2014), $T_w$ (Häggmark et al., 2000), and $T_w$-$\Gamma_{low}$ nomogram (Sims and Liu, 2015) methods. The diagnosed WPT is verified using WPT data obtained from particle size velocity (PARSIVEL) disdrometers collected during the ICE-POP 2018 period (Nov. 2017–Apr.  2018).

## 2. Data

### 2.1 ICE-POP 2018 observation sites

The northeastern region of South Korea is characterized by cold air and warm ocean temperatures in winter and complex, steeply sloped terrain from mountain ranges to the ocean (Fig. 1). An intensive observational survey was conducted in this region during the ICE-POP 2018 campaign from November 2017 to May 2018, with twenty PARSIVELs installed at 18 sites

(the cross symbols in Fig. 1a) along the coastline and in the mountain ranges to record the WPTs under various atmospheric conditions. Rawinsonde observations were also made every 3 h at five sites: two sites in the coastal region (Sokcho [SCW] and Gangwon Weather Administration [GWW]), one site in the entrance of the mountain ranges (Bokwang1-ri Community Center [BKC]), and two sites in mountain valleys (Myeonon Observatory [MOO] and Daegwallyeong Regional Weather Office [DGW]). In addition, 11 micro rain readers (MRRs) were installed at some of the sites (square symbols in Fig. 1a). The

MRRs are vertically orientated K-band radars and their data are useful for understanding the vertical precipitation characteristics of precipitation systems.

The eastern sites in the Taebaek Mountains are at a relatively low altitude, with SCW, GWW, and BKC 18, 79, and 175 m above mean sea level (MSL), whereas the western sites (DGW and MOO) are 773 and 532 m above MSL, respectively. We analyze the PARSIVEL data to identify the WPTs from the five sites (SCW, MOO, BKC, Gangneung-Wonju National

University [GWU], and DGW: Figs. 1b–1f) that are collocated with or closest to a rawinsonde observation. The PARSIVEL data at GWU are matched with the sounding data from GWW, which is about 3.88 km away with a similar altitude (GWU: 36





m MSL). This atmospheric environment and high-resolution soundings are optimal for comprehensively testing the diagnosis of the WPT.

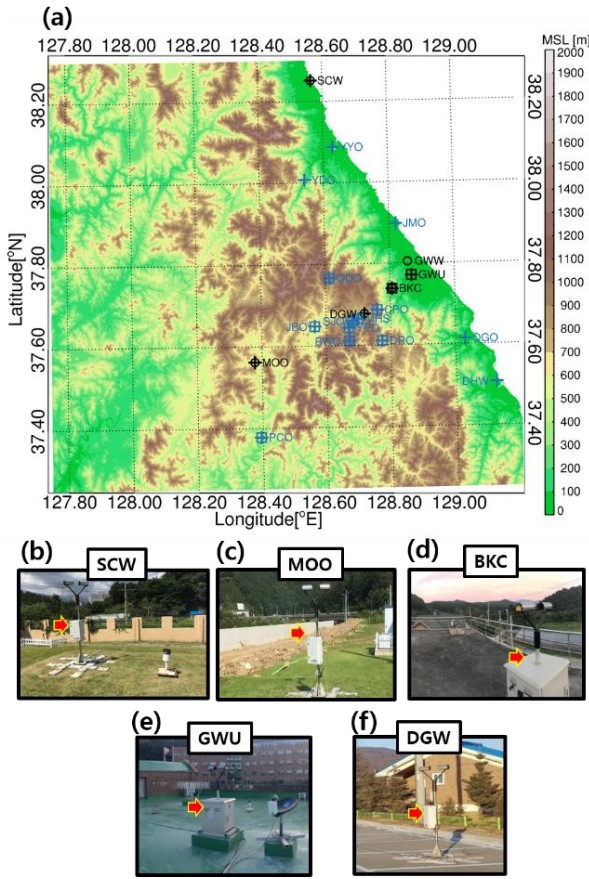

**Figure 1. (a) Topography and observational supersites in the north-east region of South Korea during the ICE-POP 2018 period. Photographs of PARSIVELs at the five sites: (b) SCW, (c) MOO, (d) BKC, (e) GWU, (f) DGW. Cross symbols indicate PARSIVELs and squares (circles) indicate MRRs (rawinsondes) in (a). The sites used in this study are labelled with text in (a).**

## 2.2 Observational data and quality control

A PARSIVEL is a disdrometer that uses a laser beam with a wavelength of 780 nm (Löffler-Mang and Joss, 2000) to obtain a particle's equivolume diameter ($D$, mm) and fall velocity ($V_f$, m s$^{-1}$) based on changes in the laser beam signals. Version 2 PARSIVELs and level 1 data are used in the present study. Level 1 data are format-converted with no processing and provide particle counts for individual diameter and velocity channels (a 32 by 32 array) every 1 min. Because the observed PARSIVEL data contain outliers that may be the result of various forms of error, such as calibration errors and "margin fallers" (Yuter et





al., 2006), we eliminate any of the level 1 data that meet one or both of the following two criteria: i) $D < 1$ mm and ii) $V_f >$ 1.4$V_a$. $V_a$ is the empirical relationship between $D$ and $V_f$ established by Atlas et al. (1973).

A modem-type rawinsonde (M10) is used for the ICE-POP 2018 campaign (In et al., 2018). The specifications for this rawinsonde are described in In et al. (2018). The observation variables recorded by the M10 rawinsonde are pressure ($P$, hPa),

$T$ (°C), RH (%), wind speed (WS, m s$^{-1}$) and wind direction (WD, °) at 1 s intervals. Additionally, $T_w$ is calculated using the two-parameter relationship for $T$ and RH suggested by Stull (2011).

The MRRs are modulated continuous wave (FMCW) radar instruments using a solid-state transmitter with a frequency of 24 GHz (Maahn and Kollias, 2012). Raw data from the MRRs are quality-controlled using de-aliasing and the noise removal algorithm suggested by Maahn and Kollias (2012). The processed MRR data are used to provide additional context for

important cases in the present study.

## 3. Methods

### 3.1 Determining the winter precipitation type

Three WPTs are considered in the present study: SN, RA, and RASN. SN is defined as solid precipitation such as dry snow,

while RA is defined as liquid precipitation. FZRA is included in RA because FZRA it is in a liquid phase when observed by a PARSIVEL. RASN is mixed-phase precipitation that includes wet snow. IPs are very difficult to identify using only PARSIVEL data and no photographic data because the $V_f$ of IP can have two modes: a low-speed mode that is similar to the $V_f$ of graupel or hail and a high-speed mode similar to raindrops (Nagumo and Fujiyoshi, 2015). Thus, the lack of multi-angle snowflake cameras (MASCs) or similar equipment at some sites (DGW, SCW, and MOO) is an issue for this analysis. In

addition, IPs in the Pyeongchang region are only observed under very specific atmospheric conditions (i.e., very strong inversion strength [> 5 K] with a freezing layer at 800–900 hPa and melting layer at 700–800 hPa; Chae et al., 2024) and are thus rare.

The 5 min PARSIVEL data are projected onto a Yuter et al. (2006) scheme that divides the data into three regions (RA, SN, and an ambiguous region; Fig. 2a) after which the number ($N$) of particles for each type is counted. The fraction of $F_{RA}$ and

$F_{SN}$ are calculated using the following equations:

$$\begin{cases} F_{RA} = 100 \ (\%) \times \frac{N_{RA}}{N_{Total}} \\ F_{SN} = 100 \ (\%) \times \frac{N_{SN}}{N_{Total}} \end{cases} \tag{1}$$

where $N_{RA}$ and $N_{SN}$ are the number of particles identified as raindrops and snow particles, respectively, and $N_{Total}$ is the number

of particles across all three regions.





We obtain a total of 131 matched precipitation cases to validate the five diagnostic methods during the ICE-POP period (1 November 2017–30 April 2018). Cases are identified that feature measurable precipitation at each of the five sounding sites. We identify precipitation cases at each site that satisfy two conditions: i) $N_{RA}+N_{SN} \geq 15$ within 5 min of the sounding start time, and ii) $-4\ °C < T_0 < 6\ °C$ and $RH_0 > 40\ \%$ at the sounding start time. Here, $T_0$ and $RH_0$ are the data recorded 1 s after the start

of the sounding, and they accurately represent the surface $T$ and $RH$ measured by the rawinsonde. Based on this hydrometeor-type classification scheme, the dominant WPT of the matched precipitation cases is determined using the newly developed algorithm with the quality-controlled 5 min PARSIVEL data (Fig. 2b).

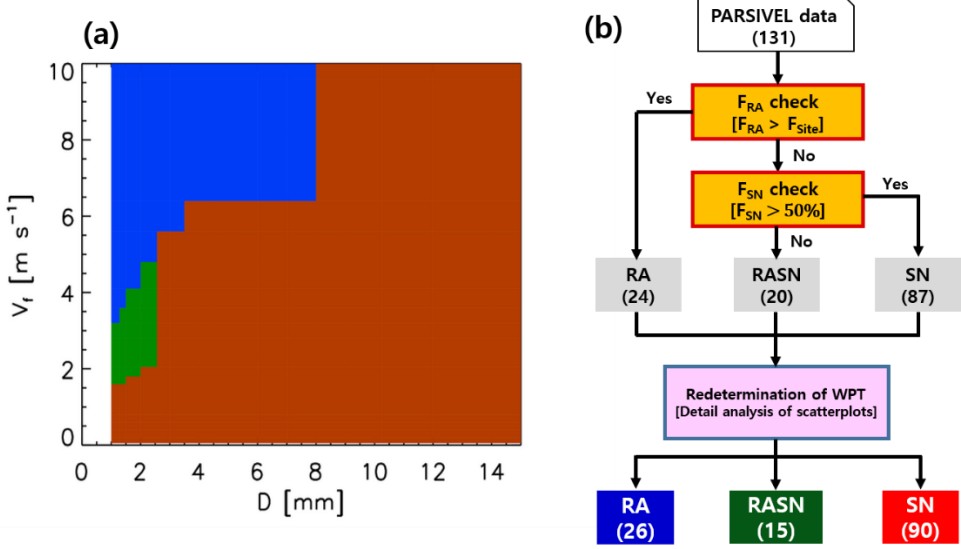

**Figure 2. (a) Yuter et al. (2006) scheme. The color indicates the determined precipitation type, where blue (red) indicates RA (SN) and green indicates ambiguous precipitation type. (b) A new decision algorithm of WPT from PARSIVEL data. The number of each WPT are shown in parentheses.**

The newly developed algorithm consists of three steps:  i) an $F_{RA}$ check, ii) an $F_{SN}$ check, and iii) redetermination of the WPT.
First, we classify RA from the matched precipitation cases while taking into account potential differences in the hardware calibration of each PARSIVEL. If the hardware is correctly calibrated, $F_{RA}$ should be 100 % for pure rainfall cases. The normalized frequency (NF) distributions of $F_{RA}$ during the ICE-POP period with a $T_{AWS}$ of > 7 °C are shown in Fig. 3. $T_{AWS}$ is the 5 min mean temperature from the nearest automatic weather station (AWS), and $F_{RA}$ is calculated using 5 min PARSIVEL data from the same site. The only exception is BKC, which has no corresponding AWS station; in this case, the  $F_{RA}$ from the
PARSIVEL at BKC and $T_{AWS}$ at GWU are matched, with the $T_{AWS}$ corrected for the difference in altitude between the two sites assuming a general temperature lapse rate (6.5 °C/km). The $F_{RA}$ at which the cumulative NF (solid black line) reaches a threshold value of 0.05 ($F_{site}$; dotted black line) varies by site (GWU: 65.2 %; BKC: 81.1 %; SCW: 61.6 %; MOO: 57.9 %;



and DGW: 68.1 %). Based on the information presented in Fig. 3, the matched precipitation cases at each site with $F_{RA} > F_{site}$ are classified as RA.

Second, we divide the remaining cases into high SN fraction ($F_{SN} > 50$ %) and low SN fraction ($F_{SN} \leq 50$ %) groups. The high SN fraction group indicates a greater likelihood of dry snowfall and is classified as SN, while the low SN fraction group indicates a greater likelihood of wet snowfall and is classified as RASN. After these first two steps, the 131 matched precipitation cases are provisionally divided into 24 RA, 20 RASN, and 87 SN cases.

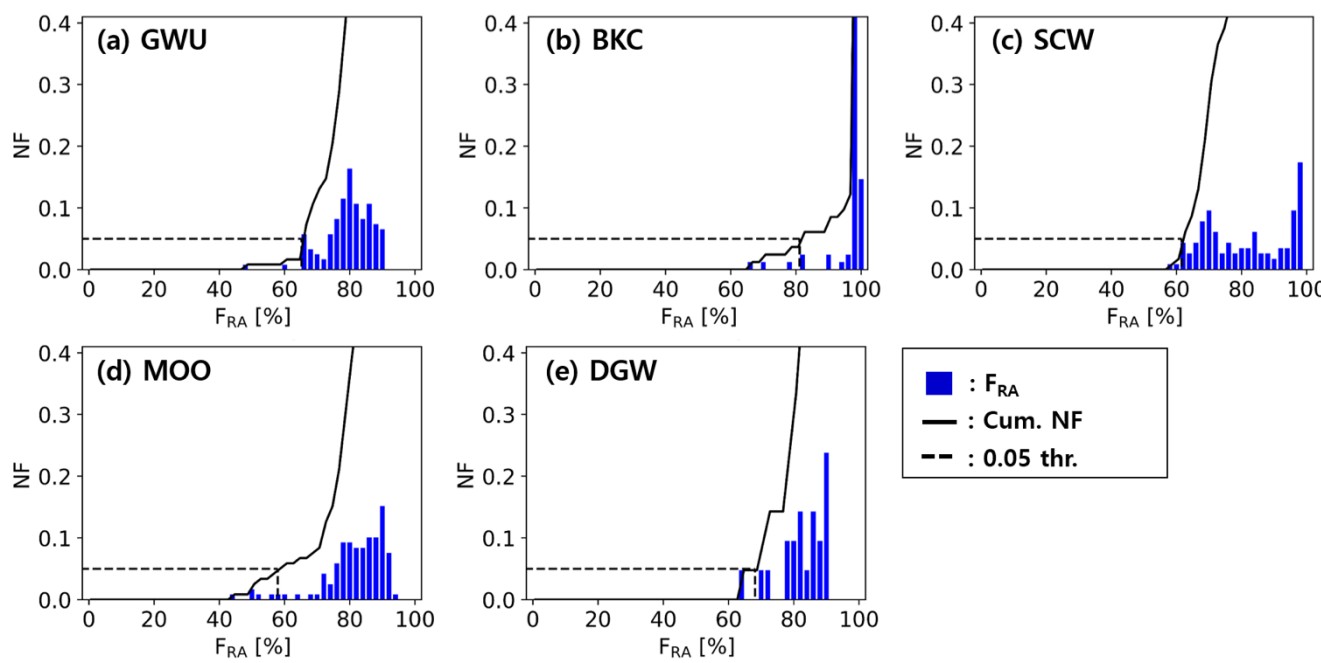

**Figure 3. Normalized frequency distribution of $F_{RA}$ of pure rainfall cases with $T_{AWS} > 7$ °C during the ICE-POP period at (a) GWU, (b) BKC, (c) SCW, (d) MOO, and (e) DGW. The black solid line is cumulative NF and the black dotted line is the 0.05 threshold.**

In the third step, we manually examine the classification results using $V_f$-$D$ scatterplots and redetermine the WPT of some

cases that are clearly misclassified. Two RA cases and an SN case that have multiple curves in the $V_f$-$D$ scatterplots are reclassified as RASN. Four RASN cases with a single curve similar to an empirical RA curve (Atlas et al., 1973) are reclassified as RA. Another four RASN cases with a single curve similar to an empirical graupel curve (Lee et al., 2015) are reclassified as SN. Two SN cases with a widely scattered distribution in their $V_f$-$D$ scatterplot despite weak wind conditions ($< 3$ m s$^{-1}$) at the near-surface are reclassified as RASN. Two RASN cases with predominantly small snowflakes with various fall speeds

are reclassified as SN because the cases are characterized by strong wind ($\geq 9$ m s$^{-1}$) near the surface and at low levels ($< 1$ km AGL), strong speed shear ($\geq 5$ m s$^{-1}$ per km), and very cold conditions (maximum $T_w$ in sounding profile $\leq -3$ °C). Strong





wind shear can lead to greater turbulence, thus generating tiny snowflakes (Dedekind et al., 2023) with chaotic movement that are more likely to be erroneously classified as RASN.

Following this redetermination step, a total of 26 RA, 15 RASN, and 90 SN cases are identified. The number of matched
precipitation cases by observation site and WPT are listed in Table 1. More than half of the SN cases (56 of 90) occur at mountain sites (DGW and MOO), whereas many of the RA cases (17 of 26) occur at coastal sites (SCW, GWU, and BKC). A similar number of RASN cases occur at both types of site.

**Table 1. The number of matched precipitation cases for each observation site and WPT.**

| Observation site | Number | SN | RASN | RA |
|---|---|---|---|---|
| SCW | 20 | 11 | 4 | 5 |
| GWU | 10 | 4 | 2 | 4 |
| BKC | 29 | 19 | 2 | 8 |
| DGW | 37 | 33 | 2 | 2 |
| MOO | 35 | 23 | 5 | 7 |
| Total | 131 | 90 | 15 | 26 |


### 3.2 winter precipitation type diagnosis methods

The SBM and four empirical methods are tested for the diagnosis of the WPT using the observed sounding data. Nomograms for the WPT for each of the four empirical methods are presented in Fig. 4. $H_{850}$ diagnoses the WPT based on a threshold
1000–850 hPa thickness that is empirically determined (Fig. 4a; Lee et al., 2014). $H_{850}$ is calculated as follows:

$$H_{850} = \frac{R_d \overline{T_v}}{g} \ln \frac{P_1}{P_2} \tag{2}$$

$$\begin{cases} \text{RA: } H_{850} \geq H_{RA} \\ \text{SN: } H_{850} < H_{SN} \\ \text{RASN: } H_{SN} \leq H_{850} < H_{RA} \end{cases} \tag{3}$$

where $R_d$ is the dry air constant (287 J K$^{-1}$ kg$^{-1}$), $g$ is the standard gravitational acceleration (m s$^{-2}$), $P_1$ is 1000 hPa, $P_2$ is 850 hPa, and $\overline{T_v}$ is the mean $T_v$ between 850 hPa and 1000 hPa. $T_v$ is calculated as a function of $T$, $P$, and RH (Lin, 2016). When the 1000 hPa data are unavailable, such as at the high-altitude sites (DGW and MOO), we use the $T_v$ at 925 hPa as an alternative for $\overline{T_v}$. The diagnosed WPT is SN if $H_{850}$ is lower than $H_{SN}$, while it is RA if $H_{850}$ is higher than $H_{RA}$. When $H_{850}$ is between $H_{SN}$ and $H_{RA}$, the diagnosed WPT is RASN. Lee et al. (2014) determined that the $H_{SN}$ and $H_{RA}$ of South Korea at low-altitude



sites (< 100 m MSL) are 1281 gpm and 1297 gpm, respectively, whereas the $H_{SN}$ and $H_{RA}$ of DGW are 1299 gpm and 1313

gpm, respectively. The WPTs at GWU, BKC, and SCW are diagnosed using the former critical values, while DGW and MOO

are diagnosed using the latter.

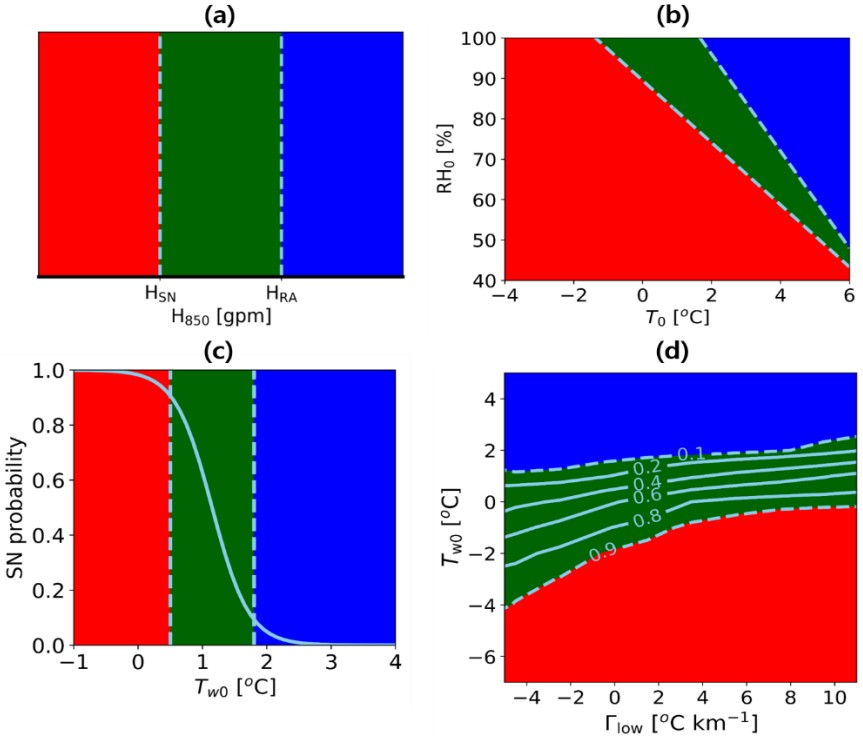

**Figure 4. Nomograms for diagnosis of WPT. The different colors indicate SN (red), RASN (green), and RA (blue). Classification of**
**WPT by using (a) $H_{850}$ and (b) $RH_0$-$T_0$ graph are shown with dashed lines suggested in Lee et al. (2014). (c) Probability function**
**(solid line) of SN as a function of the wet-bulb temperature at surface, $T_{w0}$ (Haggmark and Ivarsson, 1997). (d) Probability**
**distribution (solid and dashed lines) of SN on $T_{w0}$-$\Gamma_{low}$ graph for land areas (Sims and Liu, 2015). Here, the subscript '0' indicate the**
**near-surface value.**


The $RH_0$-$T_0$ nomogram employs the shifted Matsuo scheme suggested by Lee et al. (2014). Figure 4b presents the diagnosed

WPTs based on the $RH_0$-$T_0$ plot, with the two dashed lines derived from the following equations (Lee et al., 2014):


$$RH_0 = -12\,T_0 + 120 \tag{4}$$

$$RH_0 = -\frac{100}{13}\,T_0 + 89.5 \tag{5}$$

where $T_0$ and $RH_0$ are in °C and %, respectively. Equations (4) and (5) are used to separate RA from RASN and RASN from

SN, respectively.



Thirdly, the $T_{w0}$ method uses the probability of SN as a function of $T_{w0}$ to diagnose WPT (Fig. 4c, Häggmark et al., 2000). We used threshold probability values of 10 % and 90 % for the classification of WPT. Thus, the diagnosed WPT is SN if the wet-bulb temperature at the surface ($T_{w0}$) is lower than 0.5 °C, whereas it is classified as RA if $T_{w0}$ is larger than 1.8 °C. When 0.5 °C $\leq T_{w0} <$ 1.8 °C, the diagnosed WPT is RASN. Other probability values (20/80 % and 30/70 %) are also explored.

Using global surface-based (land station and shipboard) observations over multiple decades, Sims and Liu (2015) studied
the influence of various geophysical parameters on precipitation phases, including near-surface air $T$, atmospheric moisture, the low-level vertical $T$ lapse rate ($\Gamma_{low}$), surface skin temperature, surface pressure, and land cover type. Because snow melting occurs at close to $T_w$ (~0 °C) instead of the actual air $T$, they evaluated the SN-RA transition using $T_w$, instead of air $T$. Their analysis indicated that, in addition to $T_w$, the vertical $T$ lapse rate between the surface and 500 m significantly affects the precipitation phase. For example, at a near-surface $T_w$ of 0 °C, a lapse rate of 6 °C km$^{-1}$ results in a conditional probability of
0.86 for solid precipitation, while a lapse rate of –2 °C km$^{-1}$ (inversion) results in a probability of 0.45 (Fig. 4d: conditional probability of solid precipitation on land). Based on this finding, they developed a WPT scheme that employs $T_w$ and the low-level lapse rate as inputs and returns the conditional probability of solid precipitation. The conditional probability was derived by the ratio of the number of solid precipitation cases divided by the number of any precipitation cases under the prescribed $T_w$ and low-level lapse rate conditions. This algorithm has been incorporated into the current Global Precipitation Measurement
(GPM) mission algorithm used to determine precipitation phases (Huffman et al. 2020). Because the probability is computed using global data without accounting for regional and/or synoptic weather dependencies, its performance over the ICE-POP 2018 domain has not been examined.

Similar to the previous method, threshold probabilities of 0.1 and 0.9 are used to classify the WPT using a $T_{w0}$-$\Gamma_{low}$ nomogram (Fig. 4d), though other threshold values (0.2/0.8 and 0.3/0.7) are also explored. $\Gamma_{low}$ is the low-level lapse rate
below a height of 500 m ($\Gamma_{low}$, °C km$^{-1}$), defined as

$$\Gamma_{low} = \frac{(T_{0m\ AGL} - T_{500m\ AGL})}{0.5\ km} \qquad (6)$$

As described in Sect. 1, the SBM simulates the characteristics of precipitation particles across the size spectrum as they fall
through the ambient environment. Figure 5a presents the general process used by the SBM. When initialized from an external sounding, as followed by the present study, the cloud top is set as the maximum height with an RH of at least 80 % (Reeves et al., 2016). From the cloud top to the surface, environmental variables are then calculated and interpolated to a 10-m vertical grid spacing. Because the particle bins are independent (i.e., no aggregation/breakup is accounted for), the SBM loops through each height level for a given particle size bin before considering the next largest size. Sublimation occurs in environments
subsaturated with ice if the particles have no meltwater; if the particles do contain meltwater, evaporation occurs if the environment is subsaturated with water. Similarly, melting occurs if there is ice mass remaining and the surface $T$ of the particle reaches 0 °C. Refreezing occurs under two conditions: if there is both liquid and ice present in a particle and the $T_w$ is below 0





°C, or if the $T_w$ is at or below the nucleation $T$ ($T_c$, °C) regardless of the remaining ice mass because re-nucleation is assumed

to occur. Each microphysical process results in temporal trends in either the ice and/or water mass, which is used to calculate

the total change in ice or water mass within a given grid level based on the particle residence time. After this, all of the particle

properties (e.g., density and terminal velocity) are updated to reflect the new mass and composition of each particle, and these

serve as the initial particle properties for the subsequent grid level. This process continues until the bin is empty (i.e., the entire

particle mass has sublimated or evaporated) or until the surface is reached. For more details, see Reeves et al. (2016) and Carlin

and Ryzhkov (2019).


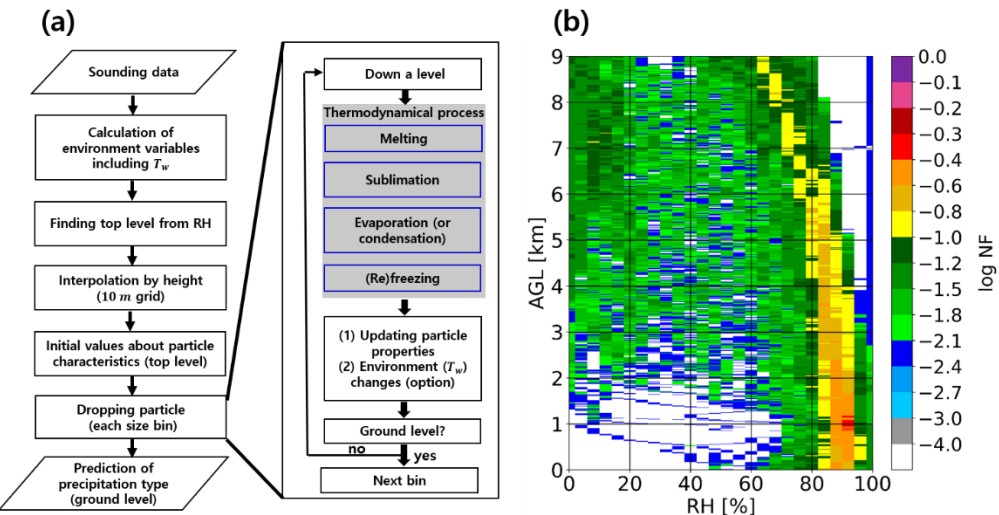

**Figure 5. (a) Flow chart describing the SBM structure (adapted from Reeves et al., 2016; Carlin and Ryzhkov, 2019). (b) Contoured frequency by altitude diagram (CFAD) of RH from rawinsonde data for the 131 matched precipitation cases.**

Once all of the particle characteristics at the surface have been calculated, an overall WPT classification is determined based

on the relative rainfall rate ($R$; mm h⁻¹) and snowfall rate ($SR$; mm h⁻¹). The original SBM WPT logic considers the relative

fractions of $R$ and $SR$, the cloud-top $T$ (to determine whether ice nucleation occurs), the number of times the $T_w$ profile crosses

0 C°, and the surface $T_w$ (Reeves et al., 2016) to determine which of the six WPTs is dominant. However, in the present study,

we are primarily interested in RA, SN, and RASN only. Therefore, we simplify the original classification scheme as follows:

290

$$\begin{cases} \text{RA: } R > 0 \text{ mm h}^{-1} \text{ and } SR = 0 \text{ mm h}^{-1} \\ \text{SN: } R = 0 \text{ mm h}^{-1} \text{ and } SR > 0 \text{ mm h}^{-1} \\ \text{RASN: } R > 0 \text{ mm h}^{-1} \text{ and } SR > 0 \text{ mm h}^{-1} \end{cases} \qquad (7)$$

The SBM parameters used in this study are presented in Table 2. The particles are separated into 20 size bins and initialized

as unrimed low-density snow aggregates. The size bins are delineated such that the equivolume diameters of fully melted





particles of equal mass in each bin are 0.1 mm apart. The largest size bin used in this study, with a fully melted equivolume diameter $D_{mw,max}$ of 1.95 mm, is about 2 times the mean value of the mass-weighted mean diameter (~ 1 mm) obtained from long-term rainfall observations in South Korea (Bang et al., 2020; Kwon et al., 2020). The $T_c$ is set to –6 °C following Reeves et al. (2016). The initial PSDs are assumed to be inverse exponential ($\mu = 0$) and are obtained through a statistical analysis of PARSIVEL data in the Pyeongchang region (Bang et al., 2019). The average values of $N_0$ and $\lambda$ (Table 2) are taken from the averages of the leeward and windward sites examined by Bang et al. (2019). Because the PSDs used to initialize the model are measured at the surface, we assume no mass growth/loss from the particles (e.g., evaporation/sublimation) for simplicity and instead only consider melting/refreezing. The assumption of mass conservation should generally be valid for this study because almost all of the precipitation cases are nearly saturated (RH > 80 %) below 5 km AGL (Fig. 5b).

**Table 2. Parameters for the SBM simulation used in this study.**

| Control variable | Value |
|---|---|
| SN habit | Aggregates |
| $r_d$ (Riming degree) | 1 (No riming) |
| Bin size and $D_{max}$ (in terms of melted diameter $D_{melt}$) | 0.1 mm, 1.95 mm |
| $T_c$ (Nucleation temperature) | -6 °C |
| $N_0$ (Intercept parameter) | 5834 m$^{-3}$ mm$^{-1}$ |
| $\lambda$ (Slope parameter) | 1.22699 mm$^{-1}$ |
| $\mu$ (Shape parameter) | 0 |
| Thermodynamic processes | Melting, (Re)freezing |

**3.3 Evaluation methods**

We evaluate the performance of the five different methods against the observed WPTs described in Sect. 3.1. We quantitatively evaluate the methods using the hit rate ($h$, %) and modified hit rate ($h'$, %) as the skill scores:

$$h = \frac{E}{O} \times 100\% \tag{8}$$

$$h' = \frac{1}{3}(h_{SN} + h_{RASN} + h_{RA}) \times 100\% \tag{9}$$





where $O$ is the number of observed cases, and $E$ is the number of correctly diagnosed cases from among the observed cases for each method. We calculate the overall $h$, $h_{SN}$, $h_{RASN}$, $h_{RA}$, and $h'$ for each of the diagnosis methods. Here, subscript of $h$ represents the accuracy for each WPT type, while $h'$ is the average accuracy across all three WPTs. The skill scores are also compared between the mountain sites (DGW and MOO) and coastal sites (GWU, SCW, and BKC), and the effect of vertical $T_w$ profiles on the accuracy of each diagnosis method is investigated to assess the strengths and weaknesses of each diagnosis method.

We also evaluate the microphysics scheme in the SBM by analyzing cases that are misdiagnosed by the SBM. Misdiagnosis of the precipitation in the Pyeongchang region may occur due to regional differences in microphysical precipitation characteristics. In particular, the original SBM uses a snow density–diameter relationship obtained from 2D-video disdrometer (2DVD) data in Colorado ($\rho_s = 0.178\, D^{-0.922}$; Brandes et al., 2007).

A region-specific density–diameter relationship is derived from 2DVD measurements at DGW (collected by Lee et al., 2015) to reflect the microphysical characteristics of snow in this region. A power-law-based regression is performed using the weighted total least square (WTLS) method (Amemiya, 1997) to minimize the deviation from both the $x$ and $y$ axes (Lee et al., 2015). The region-specific density-diameter relationship about dataset including dendrite, plate, and needle is derived as follows:

$$\rho_s = 0.09\, D^{-1.01} \tag{10}$$

In this relationship, the density of the snow particles in this region is generally lower than Brandes et al. (2007), though with a similar inverse relationship between the diameter and density. Using this relationship to optimize the microphysical scheme of the SBM, we investigate the performance of the optimized model for the misdiagnosed cases.

## 4. Results

### 4.1 Overall accuracy of the diagnosed precipitation types

We evaluate the accuracy of the $H_{850}$ method, an $RH_0$-$T_0$ nomogram, the $T_{w0}$ method with 10 % and 90 % probability values, a $T_{w0}$-$\Gamma_{low}$ nomogram with 0.1 and 0.9 threshold values, the original SBM, and the optimized SBM (Figs. 6a–f, respectively) in terms of diagnosing the WPT of the matched precipitation cases. Overall, the optimized SBM produces the highest $h$ (93.1 %) and $h'$ (87.5 %). The lowest $h$ is achieved by the $RH_0$-$T_0$ nomogram (71.8 %), while the lowest $h'$ is exhibited by the $T_{w0}$ method (68.4 %). The $H_{850}$ method ($h$: 72.5 %; $h'$: 77.5 %) and the $RH_0$-$T_0$ nomogram ($h$: 71.8 %; $h'$: 78.0 %) have similar skill scores. The skill score sensitivity of the $T_{w0}$ and $T_{w0}$-$\Gamma_{low}$ nomogram methods is analyzed according to probability values (or threshold values). The skill scores of the $T_{w0}$ method with 20 % and 80 % probability values are $h$=86.3 % and $h'$=66.6 %, compared to $h$=86.3 % and h′=63.7 % for probability values of 30 % and 70 % (data not shown), which are lower than those

for the default 10 % and 90 % thresholds (Fig. 6c). In addition, the skill scores of the $T_{w0}$-$\Gamma_{low}$ nomogram with 0.2 and 0.8 threshold values are $h$=88.5 % and $h'$=78.8 %, while that for 0.3 and 0.7 threshold values are $h$=90.8 % and $h'$=77.1 % (data not shown). The $T_{w0}$-$\Gamma_{low}$ nomogram with 0.1 and 0.9 threshold values has a lower $h$ (85.5 %) and a higher $h'$ (85.6 %) (Fig. 6d).

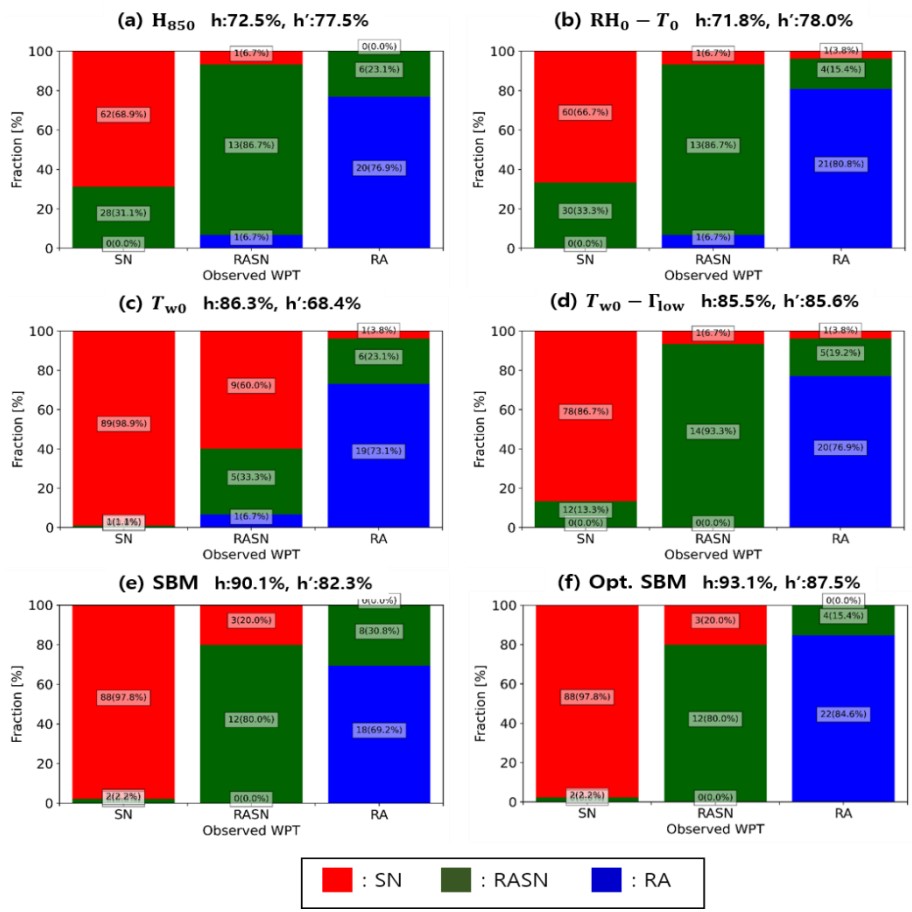

**Figure 6. Evaluation summary of the five diagnosis methods for 131 matched precipitation cases. The methods are (a) the thickness method H$_{850}$, (b) shifted Matsuo scheme on a RH$_0$ - $T_0$ nomogram, (c) the wet-bulb temperature method $T_{w0}$, (d) Sims and Liu scheme on a $T_{w0}$-$\Gamma_{low}$ nomogram, (e) the SBM, and (f) the optimized SBM. The x-axis is observed precipitation type and the colors indicate the fraction of the diagnosed precipitation types: red: SN, green: RASN, blue: RA. The diagnosed fraction of precipitation types is shown on the y-axis with the number of cases labelled in each bar. The hit rate ($h$) and modified hit rate ($h'$) are shown in the numbers on the top of each image.**

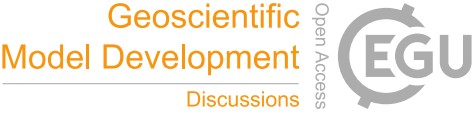

Although the $H_{850}$ and $RH_0$-$T_0$ nomogram methods are optimized for the Korea region, their $h$ and $h'$ are lower than those of the SBM and $T_{w0}$-$\Gamma_{low}$ nomogram. The $T_{w0}$ method exhibits a relatively large difference between $h$ (86.3 %) and $h'$ (68.4 %), with the inclusion of $T_{w0}$ improving the diagnosis of SN (Figs. 6c and 6d). Although the $T_{w0}$ method has the highest $h_{SN}$ (98.9 %), a significant number of RASN events are misdiagnosed as SN, reducing $h'$ and indicating that $T_{w0} = 0.5$ °C is too warm for the threshold. This is supported by the global statistics presented in Fig. 4d. The 90 % conditional probability of SN for land areas varies from $T_{w0} = -0.1$ °C at $\Gamma_{low} = 11$ °C km$^{-1}$ to $T_{w0} = -4.1$ °C at $\Gamma_{low} = -5$ °C km$^{-1}$ (Sims and Liu, 2015). The $T_{w0}$-$\Gamma_{low}$ nomogram approach has a higher accuracy for the diagnosis of RASN cases and lower accuracy for SN cases compared to the SBM. However, $h_{RA}$ is relatively low for the SBM ($h$: 69.2 %), with 8 of the 26 RA cases diagnosed as RASN, suggesting that the amount of melting differed between the simulations and the actual event. The diagnosis accuracy for the RA cases improves when using the optimized SBM by about 15 %, while the accuracy for the other WPTs does not differ between the original and optimized SBMs.

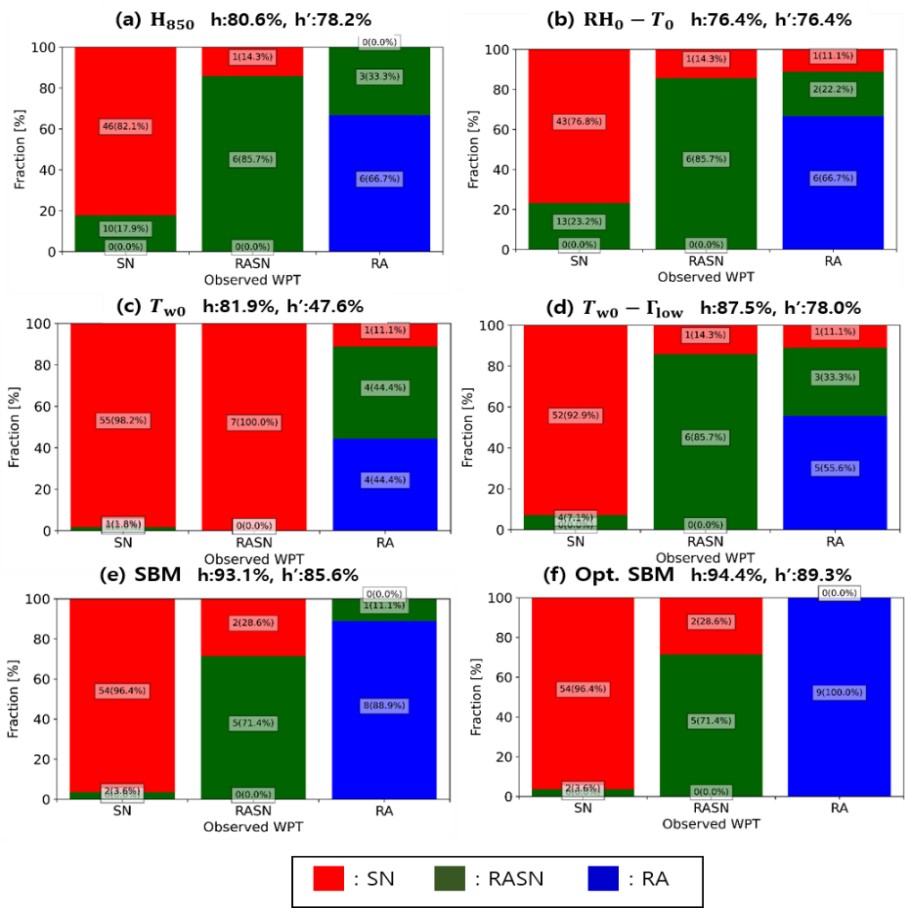

**Figure 7.** Same as in Fig. 6 except for the 72 mountain cases.





The matched precipitation cases are divided into mountain sites (DGW and MOO) and coastal sites (GWU, BKC, and SCW), with the skill scores presented in Figs. 7 and 8, respectively. For the mountain sites, the $h$ (94.4 %) and $h'$ (89.3 %) are the highest for the optimized SBM (94.4 % and 89.3 %, respectively) and the lowest values are observed with the $RH_0$-$T_0$ nomogram approach ($h$: 76.4 %) and the $T_{w0}$ method ($h'$: 47.6 %; $h_{RASN}$: 0 %). For the coastal sites, the $T_{w0}$ method and the optimized SBM ($h$: 91.5 %) and the $T_{w0}$-$\Gamma_{low}$ nomogram method ($h'$: 88.2 %) exhibit the highest accuracy; in contrast, the $H_{850}$ produces the lowest accuracy ($h$: 62.7 %; $h'$: 72.3 %; $h_{SN}$: 47.1 %). The skill scores for the $T_{w0}$ method at the mountain sites are lower than the coastal sites, whereas the opposite is true for the $H_{850}$ method and the $RH_0$-$T_0$ nomogram approach. In contrast, the skill scores for the SBM are higher at the mountain sites compared to coastal sites, while the skill scores for the optimized SBM with RA cases are higher than those for the original SBM at both mountain sites and coastal sites. The $h$ of the $T_{w0}$-$\Gamma_{low}$ nomogram at the mountain sites is higher than the $h$ at the coastal sites, while the $h'$ is lower.

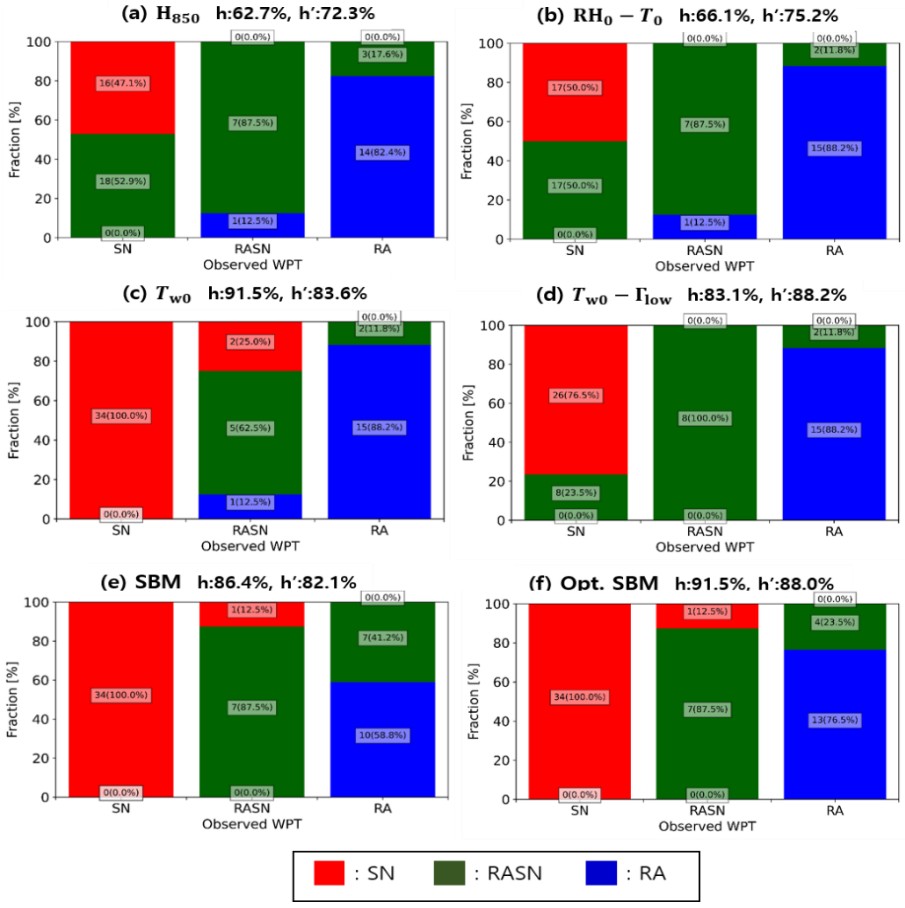

**Figure 8. Same as in Fig. 6 except for the 59 coastal cases.**





The dependence of the skill scores on the terrain for all five methods is also explored. The $H_{850}$ and $RH_0$-$T_0$ nomogram methods exhibit large differences in $h_{SN}$ with changes in the terrain, with the mountain sites scoring higher than coastal sites. The $T_{w0}$ method also exhibits significant differences in both $h_{RASN}$ and $h_{RA}$ with terrain, with coastal site scores exceeding mountain sites. The $T_{w0}$-$\Gamma_{low}$ nomogram method and the SBM exhibit considerable differences in $h_{RA}$, with the $T_{w0}$-$\Gamma_{low}$

nomogram producing a lower $h_{RA}$ at the mountain sites than the coastal sites and the SBM demonstrating the opposite.

### 4.2 Dependence of diagnosis accuracy on wet-bulb temperature profiles

The environments of the coastal and mountain sites in the Pyeongchang region differ in many respects. In the coastal region, the low-level atmosphere is more humid and warmer than the mountain region due to the East Sea. The mountain region often

has ILs near the surface due to radiational cooling, regional subsidence inversion, and cold-air damming. Near-surface ILs can strongly influence the accuracy of ground-based WPT diagnosis. The difference in the altitudes between the two regions also affects $V_f$ because of differences in air density. Wind shear effects associated with specific synoptic wind patterns can enhance riming processes in the mountains of the Pyeongchang region (Kim et al., 2021). Therefore, we assess the impact of the atmospheric conditions on the performance of the five methods based on the characteristics of $T_w$ profiles.

Figure 9 presents the observed WPTs based on the nomograms used for the four empirical methods. Figure 9a shows the distribution of observed WPTs using the $H_{850}$ method. The $H_{850}$ values for SN cases range from 1273 gpm to 1305 gpm at mountain sites and from 1269 gpm to 1297 gpm at coastal sites. In contrast, the $H_{850}$ values for RA cases range from 1297 gpm to 1329 gpm at mountain sites and from 1289 gpm to 1321 gpm at coastal sites. For RASN, a large proportion of the $H_{850}$ values are distributed between $H_{SN}$ and $H_{RA}$, as suggested by Lee et al. (2014), at both mountain and coastal sites. However,

the $H_{850}$ of many SN cases overlaps with that of RASN cases, with the overlap especially noticeable at coastal sites. The $H_{850}$ values for RASN cases range from 1294 gpm to 1308 gpm at the mountain sites and from 1282 gpm to 1302 gpm at the coastal sites.

Figure 9b presents $RH_0$-$T_0$ scatterplots with the shifted Matsuo scheme (Lee et al., 2014). Many SN cases are misdiagnosed as RASN due to the low $T_0$ threshold value when $RH_0 > 85$ %. Thus, we can speculate that the advection of low-level warm

and humid air ($T_0 = \sim 0$ °C; $RH_0 > 85$ %) during snow is likely to increase its misdiagnosis as RASN.

Figure 9c displays the distribution of observed WPTs using $T_{w0}$. The dashed lines at $T_{w0} = 0.5$ °C and $T_{w0} = 1.8$ °C represent the thresholds suggested by Häggmark et al. (2000). The $T_{w0}$ of SN cases ranges from –6 °C to 1 °C and that of RA cases ranges from –1 °C to 3.5 °C for mountain sites, compared to –6 °C to 0 °C and 1 °C to 5.5 °C for coastal sites, respectively. The $T_{w0}$ of SN cases ranges from –6 °C to 1 °C and that of RA cases ranges from –1 °C to 3.5 °C for mountain sites, compared

to –6 °C to 0 °C and 1 °C to 5.5 °C for coastal sites, respectively. The $T_{w0}$ for cases at the mountain sites ranges from –2 °C to 0.5 °C. A broad overlap of RASN and other WPTs highlights the difficulty in diagnosing WPTs in mountain regions with a single $T_{w0}$ threshold. In contrast, the distributions of WPTs as a function of $T_{w0}$ are much more clearly separated at the coastal sites.



Figure 9d presents the two-dimensional distribution of observed WPTs based on the $T_{w0}$–$\Gamma_{low}$ nomogram with a threshold

snow probability of 0.1 and 0.9 (Sims and Liu, 2015). The $\Gamma_{low}$ of RA varies widely, though it tends toward negative values.

Three RA cases with $T_{w0} > 0$ °C and $\Gamma_{low} < -2$ °C km$^{-1}$ have a ground IL at the mountain sites and two of them are misdiagnosed

as RASN. In addition, two mountain cases are misdiagnosed as RA with high $\Gamma_{low}$ (> 8 °C km$^{-1}$). One is possibly FZRA ($T_{w0}$

~ –1 °C) and the other ($T_{w0}$: ~ 1 °C) indicates the presence of a complex atmospheric vertical structure. A cold RASN case

with a $T_{w0}$ of around –2 °C and a $\Gamma_{low}$ of ~ 8.5 °C km$^{-1}$ also requires investigation into the atmospheric vertical structure.


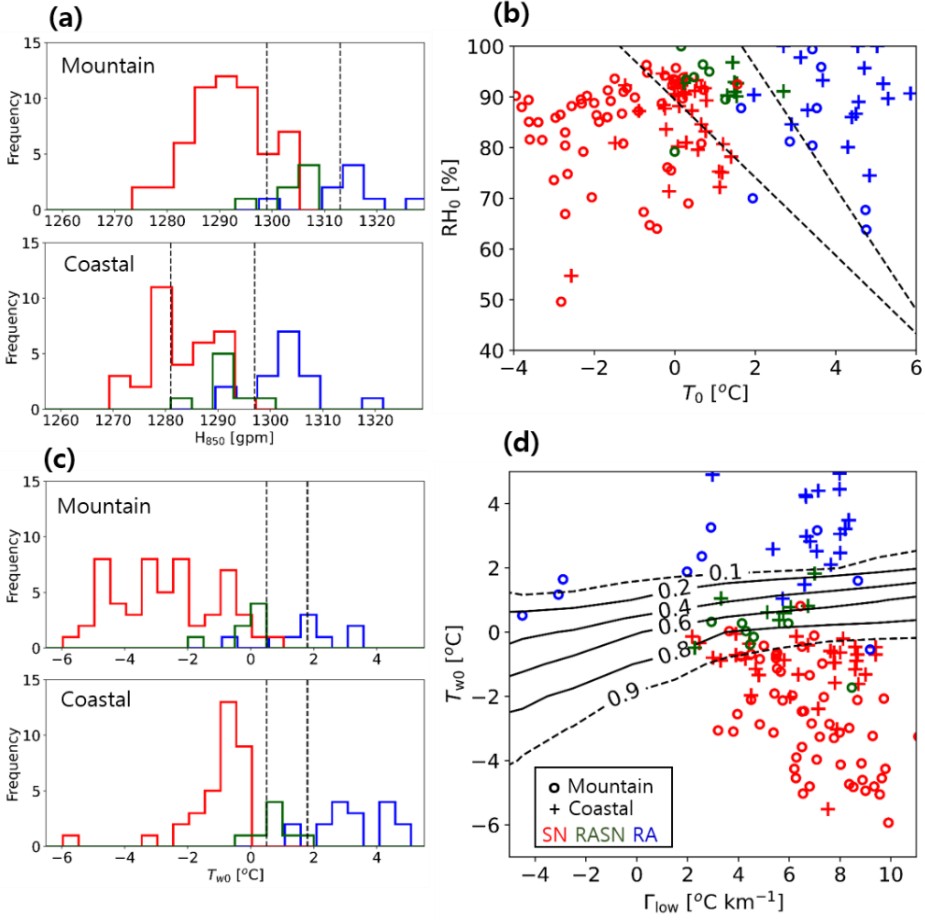

**Figure 9. Representation of observed WPTs (colors) on (a) $H_{850}$ histogram, (b) $RH_0$-$T_0$ scatterplots, (c) $T_{w0}$ histogram, and (d) $T_{w0}$-$\Gamma_{low}$ scatterplots. The colors indicate observed WPTs: SN (red), RASN (green) and RA (blue). The circle (cross) symbols indicate mountain (coastal) sites. The dashed lines indicate the threshold values for diagnosing WPTs.**


The performance of each diagnosis method is also investigated as a function of the atmospheric vertical structure (i.e., the

$T_w$ profile) (Fig. 10–12). Figures 10a and 10b display the $T_w$ profiles for observed SN cases at mountain and coastal sites,

respectively, with bold lines indicating misdiagnosed cases. Figure 10 shows that characteristics of the $T_w$ profile below 1 km



AGL strongly influence the performance of all five diagnosis methods for SN cases. The $H_{850}$ and $RH_0$-$T_0$ nomogram methods
tend to misdiagnose SN as RASN when relatively warm conditions are present below 1 km AGL. This tendency is especially
noticeable at coastal sites, suggesting that SN cases with relatively warm and moist environments are frequently observed at
coastal sites in the Pyeongchang region. These cases can be accurately diagnosed as SN by using $T_{w0}$ as the threshold instead
of $T_0$. The $T_{w0}$-$\Gamma_{low}$ nomogram tends to misdiagnose the WPT in some warm environments with low vertical lapse rates that
occur at coastal sites, indicating that a slight adjustment of the $T_{w0}$ threshold is required. The two SN cases misdiagnosed by
the SBM at the mountain sites have a very thin (< 50 m depth) near-surface warm layer (WL; a layer with $T_w > 0$ °C in this
study), highlighting a potential need to slightly increase the wet-bulb temperature threshold used to partition SN from RASN
for very shallow near-surface WLs.

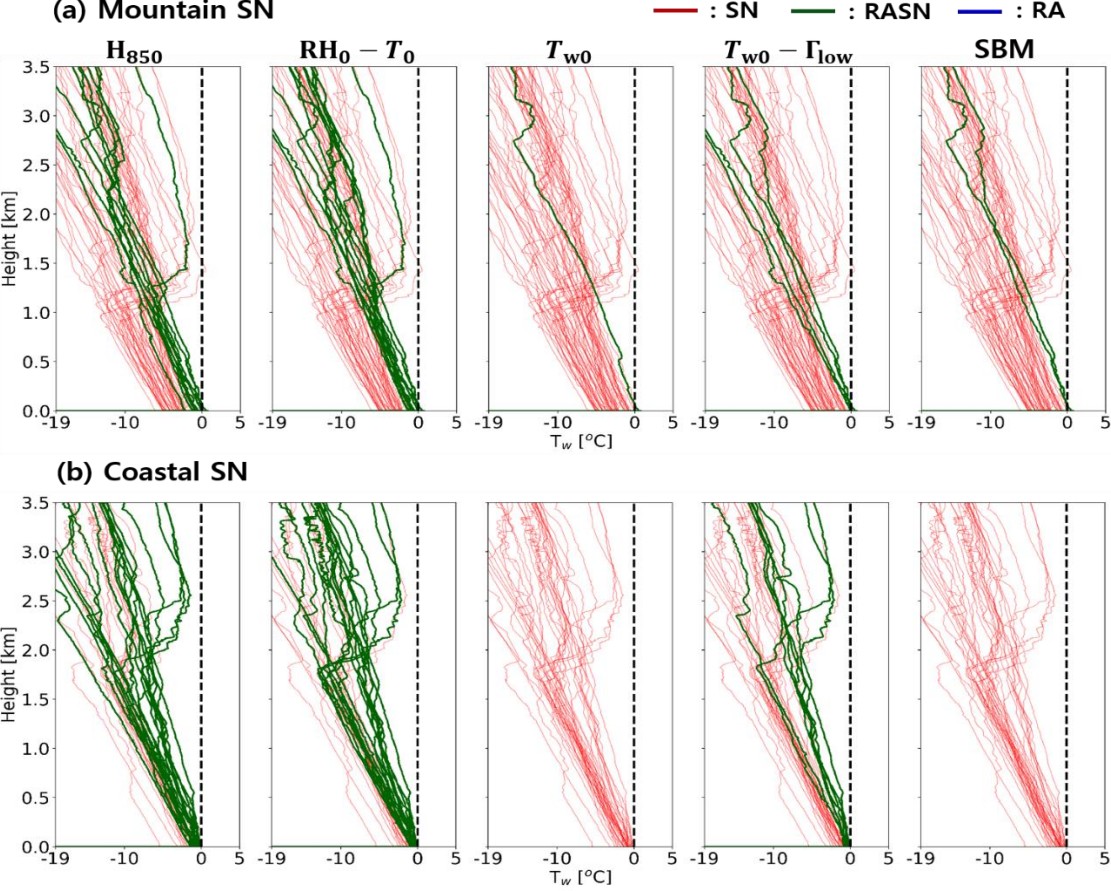

**Figure 10.** $T_w$ profiles for observed SN cases occurring at (a) mountain sites and (b) coastal sites. The blue, red, and green lines
indicate diagnosed RA, SN, and RASN cases, respectively, for the $H_{850}$ thickness, shifted Matsuo scheme on $RH_0$ - $T_0$ nomogram,
wet-bulb temperature $T_{w0}$, $T_{w0}$-$\Gamma_{low}$ nomogram, and the SBM methods. Bold lines indicate misdiagnosed cases.



Figures 11a and 11b present the $T_w$ profiles of the mountain and coastal sites, respectively, for RASN cases. The ground
temperature for mountain RASN cases tends to be colder than that for coastal RASN cases, which strongly influences the
performance of the wet-bulb temperature method. An RASN case at the mountain site MOO (21 UTC on 4 Mar 2018; the
same as the cold RASN case with a $T_{w0}$ of approximately –2 °C and a $\Gamma_{low}$ of ~ 8.5 °C km$^{-1}$) is misdiagnosed by all five methods.
The $T_w$ profile of this case has an isothermal layer at 1.7–2.2 km AGL with a $T_w$ of about –0.5 °C. However, the PARSIVEL
data clearly reveals the presence of liquid-phase particles ($F_{RA}$ = 42.55 %). We assume that melting occurs in the isothermal
layer although there are no data revealing the presence of a WL, such as vertical-pointing radar data, at MOO. The SBM
misdiagnoses two mountain RASN cases and a coastal RASN case as SN. These cases have a near-surface $T_w$ very close to
0 °C but slightly less than 0 °C.

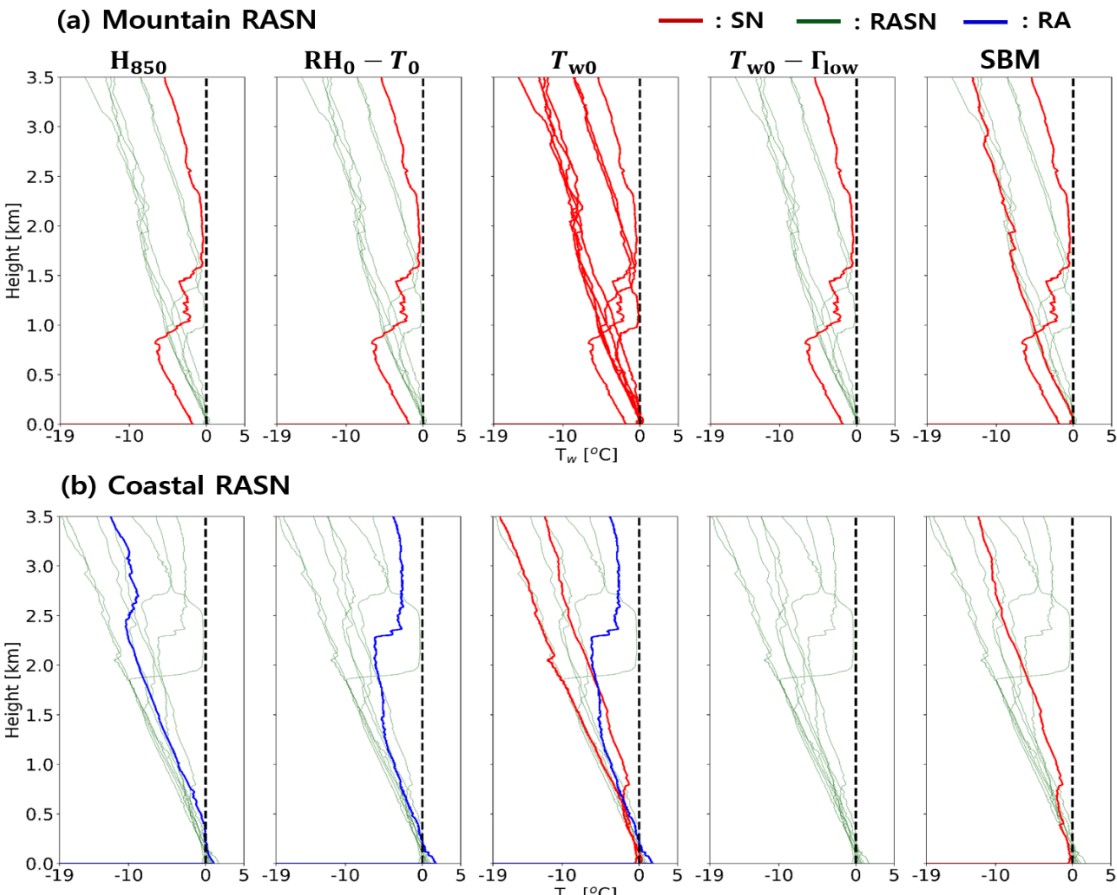

**Figure 11.** As in Figure 10, but for observed RASN cases.



Figures 12a and 12b present the $T_w$ profiles for mountain and coastal sites, respectively, for RA cases. Many RA cases at the mountain sites have a deep WL and an IL below 1.5 km AGL (the black dotted oval in Fig. 12a) whereas RA cases with a shallow WL and no IL frequently occur at the coastal sites (the black dotted oval in Fig. 12b). The SBM performs well in the

470     former scenario but poorly in the latter. Other methods produce the opposite results, with superior performance at the coastal sites. The presence of ILs makes diagnosis based solely on ground conditions difficult. The SBM simulations sufficiently melt all particles within deep WLs (i.e., 500–1500 m) in the former scenario. However, some cases of misdiagnosed RA have relatively shallow WL depths (< 500 m). The SBM simulations diagnose RASN for these cases because of the incomplete melting of large particles. For some cases with complex atmospheric profiles, the SBM diagnoses RASN for FZRA-like cases

475     (the red arrow in Fig. 12a) with a single WL and single cold layer (CL; a layer with $T_w < 0$ °C below the WL in the present study) and for two cases with double WLs and a single CL (the red arrow in Fig. 12b).

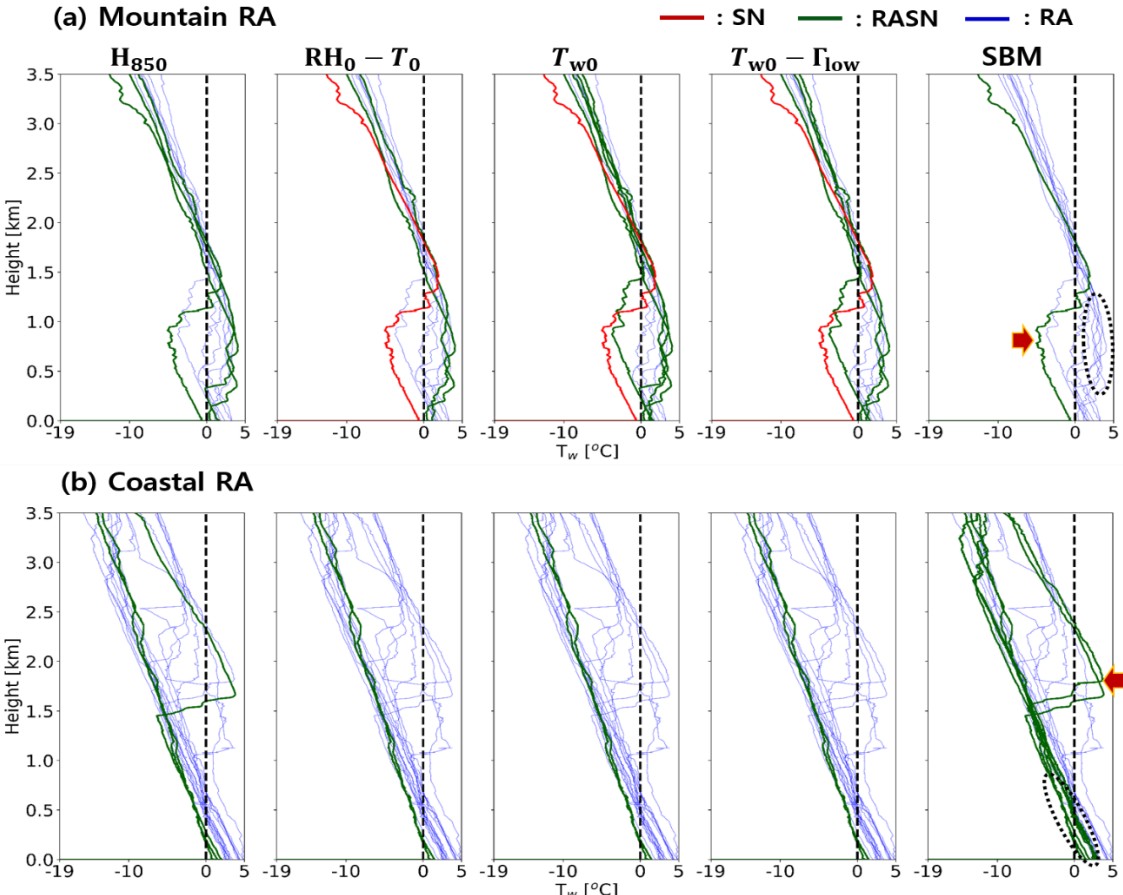

**Figure 12. As in Figure 10, but for observed RA cases. The arrows denote misclassified profiles that are discussed further in section**
480     **4.2.**





### 4.3 Analysis of the misdiagnosed cases and optimization of the Spectral Bin Model

Overall, the SBM misdiagnoses two observed SN cases, three observed RASN cases, and eight observed RA cases. The misdiagnosed SN cases have a very shallow WL near the ground. The misdiagnosed RASN cases have no WL in the $T_w$ profile but the maximum $T_w$ in the sounding profile is very close to 0 °C. We hypothesize that observation or representativeness errors in the sounding may play an important role in the misdiagnosis of these SN and RASN cases. For example, the rawinsonde sensor used in the ICE-POP project has an accuracy error of ~ 0.3 °C for $T$ (In et al., 2018). Changes in the rawinsonde path due to variation in wind speed and/or direction also can influence the diagnosis of the WPT because the atmosphere is not homogeneous.

The eight misdiagnosed RA cases are listed in Table 3. WL depth is defined as the depth of the layer with $T_w > 0$ °C in the $T_w$ profile. We divide the cases into three groups according to the WL depth and the number of WLs: (1) a single WL with a depth of more than 400 m, (2) a single WL with a depth of less than 400 m and low-level warm advection, or (3) double WLs. Group 1 has a WL with a depth of 400–600 m, Group 2 has a WL with a depth of 200–400 m and southerly flow at low levels, and Group 3 has a CL between a surface WL and a higher WL.

**Table 3. The description of RA cases misdiagnosed by the SBM. Here, WL depth is defined as the depth of the $T_w > 0$ °C layer in the profile. 'Aloft' indicates that the layer is not adjoined to the surface.**

| Group name | Date/Time | Site | WL depth | AGL with $T_w = 0$ °C |
|---|---|---|---|---|
| (1) Single WL with depth > 400 m | 28 Feb  2018 / 09 UTC | BKC | 480 m | 480 m |
| | 28 Feb  2018 / 12 UTC | BKC | 440 m | 440 m |
| | 28 Feb  2018 / 12 UTC | GWU | 410 m | 410 m |
| | 15 Mar  2018 / 15 UTC | MOO | 660 m (aloft) | 1810 m, 1150 m |
| (2) Single WL with depth < 400 m and low-level warm advection | 07 Mar 2018 / 12 UTC | GWU | 200 m | 200 m |
| | 07 Mar 2018 / 15 UTC | SCW | 170 m | 170 m |
| (3) Double WL | 15 Mar  2018 / 12 UTC | GWU | 660 m (aloft), 470 m | 2400 m, 1740 m, 470 m |
| | 15 Mar  2018 / 12 UTC | BKC | 730 m (aloft), 280 m | 2330 m, 1600 m, 280 m |

A representative example from each group is analyzed. We also compare the simulation results between the original and optimized SBMs. Figure 13 presents the environmental profiles from the rawinsondes and $V_f$–$D$ scatterplots from the corresponding PARSIVEL taken at around 1200 UTC 28 Feb 2018 from GWU in Group 1 (Fig. 13a,d), 1200 UTC 7 Mar 2018 from GWU in Group 2 (Fig. 13b,e), and 1200 UTC 15 Mar 2018 from GWU in Group 3 (Fig. 13c,f). A wide distribution of raindrop sizes ($D_{max}$: ~ 4.25 mm) following the RA curve is presented in Fig. 13d for a 400-m WL with strong easterly winds





(Fig. 13a). In contrast, a narrow raindrop size distribution ($D_{max}$: ~ 1.62 mm) following the RA curve is displayed in Fig. 13e for a 200-m WL with strong low-level southerly winds (Fig. 13b). Fig. 13f also shows a relatively narrow raindrop size distribution ($D_{max}$: ~ 2.5 mm) with the matched profile characterized by an elevated WL and another WL near the ground (Fig. 13c).

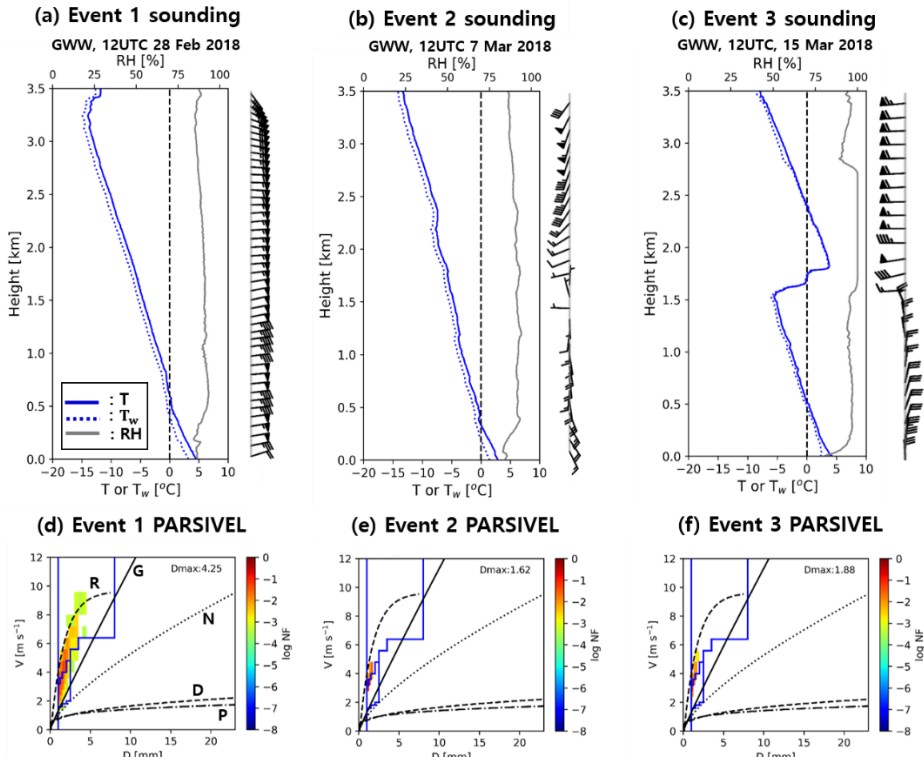


**Figure 13.** Environmental profiles (a-c) from rawinsondes and (d-f) $V_f$ - $D$ scatterplots from PARSIVELs for Event 1, Event 2 and Event 3, respectively. The blue solid (dotted) line in environmental profile indicates $T$ ($T_w$) and the grey solid line is RH. 'P' (Plate), 'D' (Dendrite), 'G' (Graupel), and 'N' (Needle) in the scatterplots depict empirical size-fall speed relationships suggested by Lee et al. (2015). 'R' (Raindrop) is the empirical relationship suggested by Atlas et al. (1973). The Yuter et al. (2006) scheme is marked as blue solid line in the scatterplots.

Figure 14 presents the relationship between the height and liquid water fraction ($f_w$) for the original and optimized SBMs as a function of the particle size for the cases shown in Fig. 13. The simulation results from the original SBM for Event 1 show

the incomplete melting of large particles (> 1.75 mm) (Fig. 14a), whereas the $f_w$ distribution from the optimized SBM shows complete melting of all particles at 200 m AGL (Fig. 14e). The simulation results for Event 2 show that particles with a $D_w$ of ≤ 1.05 mm completely melt in the original SBM, compared with 1.35 mm for the optimized version (Figs. 14b and 14f); the optimized SBM simulation significantly increases the amount of melting. However, the maximum diameter with complete melting (~ 1.35 mm) in the simulation is still slightly smaller than the observed $D_{max}$ (~ 1.62 mm). This difference could be




the result of three sources of error: northward advection of the rawinsonde due to low-level southerly winds, hardware calibration issues for the GWU PARSIVEL, and/or the growth of raindrops via a collision–coalescence process, which is not accounted for in either SBM.

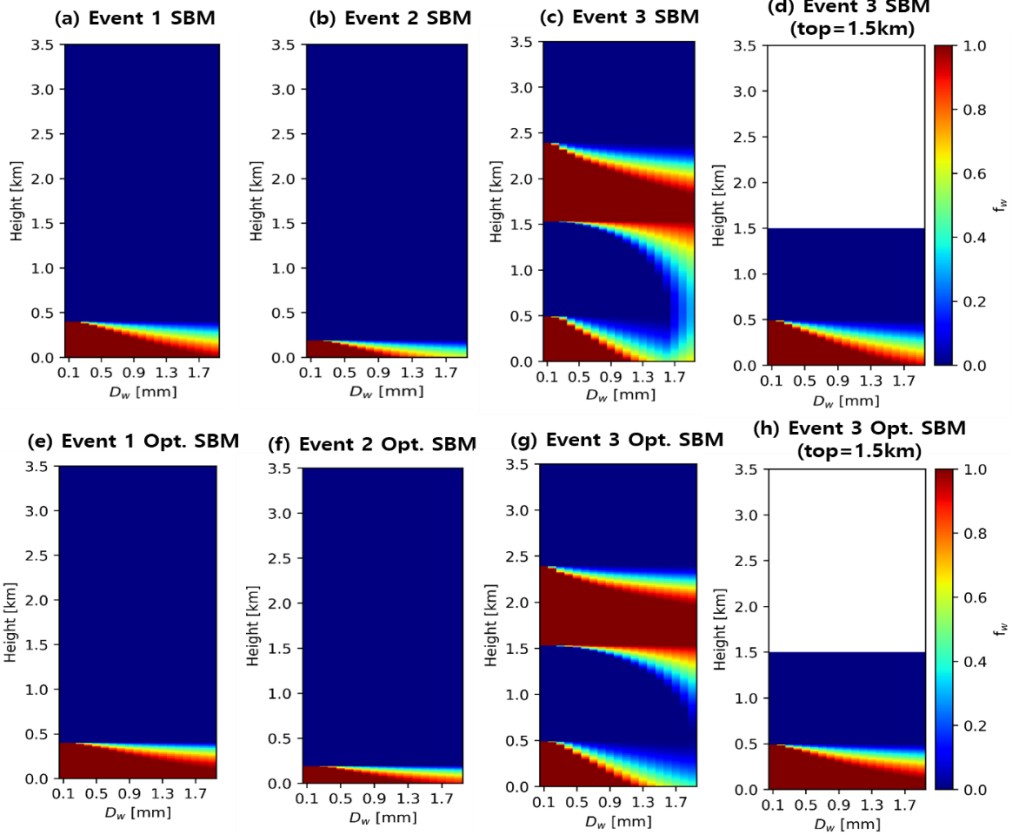


**Figure 14. Liquid water fraction distribution as a function of height and $D_w$ for Event 1, Event 2, and Event 3. (a)~(d) Simulation result of the SBM, (e)~(g) Simulation result of the optimized SBM.**

    Simulation results for Event 3 (Figs. 14c and 14g) show melting, refreezing, and additional melting during the descent of

the particles from the cloud top to the ground. At the surface, the melting of large particles is incomplete in both the original SBM and optimized SBM although a deep WL (depth: ~ 500 m) below the CL is present. The SBM assumes that the fall speed of the particles undergoing refreezing follows the relationship for IPs suggested by Kumjian et al. (2012). Because IPs generally have a larger density and fall velocity than snowflakes, the melting speed for IPs is relatively slow. To better understand Event 3, we analyzed MRR data. Figure 15 presents the time–height series for radar reflectivity (Z, dBZ) and

Doppler velocity ($-V_r$, m s$^{-1}$) observed using the MRR at GWW on 15 Mar 2018. Near the sounding time, the precipitation system drastically changes from a shallow system (cloud top of ~ 1.5 km AGL) to a seeder-feeder system (cloud top of ~ 3.5

km AGL) (Figs. 15c and 15d). It is possible that the rawinsonde sensor passed the feeder–seeder system but the ground precipitation observed by the PARSIVEL appears to originate from the shallow system observed earlier in the time series. Indeed, the Doppler fall velocities measured by the MRR from the sounding time onward between 1.0 and 1.5 km AGL are

relatively slow and are unlikely to correspond to IPs as suggested by the SBM when initialized with a higher cloud top (Fig. 14c,g). Therefore, we re-run the SBM simulation with the cloud top set to 1.5 km. The simulation results from this new run are shown in Figs. 14d and 14h. Both the SBM and the optimized SBM show an increase in melting amount in the WL near the ground compared to Figs. 14c and 14g, with the optimized SBM simulating the complete melting of all particles by 200 m AGL.


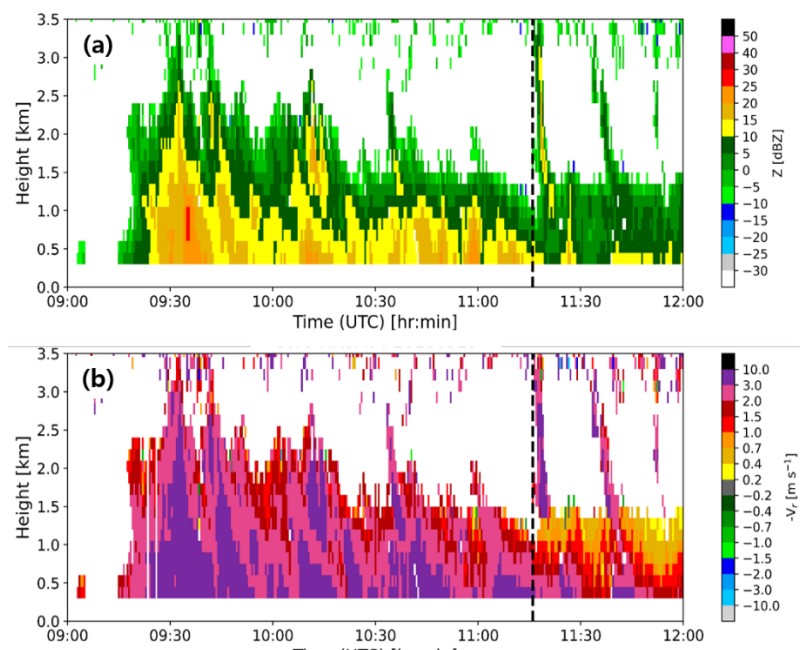

**Figure 15.** (a) *Z* and (b) -*V*$_r$ timeseries observed from the MRR in GWW on 15 Mar 2018. Solid dotted line indicates sounding start time of Event 3.

In summary, the potential main causes of the misdiagnosis of RA cases when using the SBM are suboptimal microphysical assumptions and sources of error in the input data. Optimization of the microphysics scheme using data from the Pyeongchang region significantly increases the amount of melting compared to simulations using the original microphysics scheme in the SBM. Of the eight misdiagnosed cases in the SBM, four are correctly diagnosed by the optimized SBM. If a more accurate cloud top is also considered, two more cases are correctly diagnosed by the optimized SBM. These results indicate that using

accurate cloud top information can produce more reasonable SBM simulations. Although Group 2 cases are still misdiagnosed





by the optimized SBM, the simulation accuracy could be further improved if information on horizontal advection and the maximum particle size are considered.

## 5. Summary and future work

The performance of the SBM in diagnosing WPT was evaluated through a comparison with other empirical/statistical methods (the thickness method, the shifted Matsuo scheme, the wet-bulb temperature method, and the Sims and Liu scheme) for 131 matched precipitation cases during the ICE-POP 2018 period. The observed WPTs were determined from 5 min PARSIVEL data using a newly designed decision algorithm. This algorithm classified the three WPTs SN, RASN, and RA using $F_{RA}$ and $F_{SN}$ based on the Yuter et al. (2006) scheme and manual analysis of $V_f$-$D$ scatterplots. The WPTs diagnosed by

the five methods were obtained using matched sounding data. A simplified WPT classification scheme for the SBM using $R$ and $SR$ was used, even though the SBM can classify additional WPTs. Five skill scores ($h$, $h´$, $h_{SN}$, $h_{RASN}$, and $h_{RA}$) were calculated to evaluate the performance of the diagnosis methods. In addition to the overall skill scores, the effect of the WPTs (SN, RASN, and RA), terrain (i.e., mountain vs. coastal sites), and atmospheric vertical structure ($T_w$ profiles) on the performance of the compared methods was examined.

The SBM (which ranked 1st for $h$) and the $T_{w0}$-$\Gamma_{low}$ nomogram approach (which ranked 1st for $h'$) achieved higher scores than the other methods for all matched precipitation cases. The accuracy of the SBM was highest for the mountain sites, whereas the accuracy of the $T_{w0}$ and $T_{w0}$-$\Gamma_{low}$ nomogram methods was highest for the coastal sites. Coastal SN cases featuring relatively warm and moist environments can lead to misdiagnosis when using the $H_{850}$ and $RH_0$-$T_0$ nomogram methods. Most of the RASN cases that occurred at the mountain sites were characterized by a very shallow WL near the surface. These cases

led to poor diagnosis using the wet-bulb temperature method. Ground-based or low-level-based methods showed low accuracy for mountain RA cases with a near-ground IL, whereas the SBM performed well for these cases. Conversely, the SBM exhibited relatively poor accuracy for some coastal RA cases with a WL depth of less than 500 m. These results suggested that SBM simulations tend to produce relatively low levels of melting compared to the observed precipitation.

The microphysics scheme used in the SBM was evaluated by analyzing three groups of misdiagnosed RA cases: those with

a single WL with a depth of > 400 m, those with a single WL with a depth of < 400 m and low-level warm advection, and those with double WLs. We also attempted to optimize the microphysics scheme of the SBM using a region-specific density–diameter relationship and compare the simulations between the original and optimized SBMs for the three groups. Overall, the optimized SBM demonstrated an increased amount of melting and higher skill scores than the original SBM. The optimized SBM also correctly diagnosed the WPT of the double WL group when more representative cloud top height data were used.

The potential of the SBM for diagnosing the WPT was thus confirmed in the present study. The performance of the original SBM was superior to existing optimized methods (the $H_{850}$ and $RH_0$-$T_0$ nomogram methods) and the skill scores were improved further via regional optimization of the SBM's microphysics scheme. Future work should focus on the development of a combined SBM with other reanalysis field data for the acquisition of three-dimensional WPT information.





**Code and data availability.** Code and data are available upon request (Wonbae Bang via wonbaebang@knu.kr).


**Author contributions**. WB designed and conducted the research under the supervision of GWL. JC gave advice regarding operation and troubleshooting of SBM source code. WB and KK operated PARSIVEL and processed PARSIVEL data. KK derived density-diameter relationship in Pyeongchang. WB analyze data and JC, AR, KK, GL, GWL contributed to the scientific discussions and gave constructive advice. WB wrote the manuscript with substantial contributions from all co–author.


**Competing interests**. The authors declare that they have no conflict of interest.

**Acknowledgements**. The authors greatly appreciate the participants in the World Weather Research Programme Research Development Project and Forecast Demonstration Project, International Collaborative Experiments for Pyeongchang 2018
Olympic and Paralympic winter games (ICE–POP2018) hosted by Korea Meteorological Administration. This work was funded by the Korea Meteorological Administration Research and Development Program under Grant KMI2022-00310. Additionally, funding was provided by NOAA/Office of Oceanic and Atmospheric Research under NOAA-University of Oklahoma Cooperative Agreement #NA21OAR4320204, U.S. Department of Commerce.

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
