# Peer review of "Diagnosis of winter precipitation types using Spectral Bin Model (version 1DSBM-19M): Comparison of five methods using ICE-POP 2018 field experiment data"

_Geoscientific Model Development, 2024_

## Community Comment (CC2)

**Response to Reviewer**

**19 December 2024**

**Overview:**

**The authors evaluated five diagnosis schemes of identifying winter precipitation types using data from the ICE-POP 2018 field experiment. They found that the scheme using one-dimensional spectral bin model (SBM) with the climatological snow density-diameter relationship for the Pyeongchang region demonstrates superior performance. The manuscript is well written, clear, and easy to follow. I have only some minor comments regarding clarification or justification for consideration.**

Dear Referee,

We greatly appreciate your positive feedback and the time and effort you devoted to reviewing our manuscript and dataset. We have carefully reviewed your comments and actively reflected them. Thank you again for your help in improving our manuscript.

Best regards,

Wonbae Bang (on behalf of the author team)

**Specific Comments:**

**1. Section 2.2: How large uncertainties of these observations? The authors should discuss them to enhance the manuscript's robustness.**

➔ Thank your advice and I reflect it. I add explanation about measurement errors of PARSIVEL and sounding. Also, specification of MRR is more detailed at Section 2.2.

(red color letter between 130~150 lines of revision file)

**2.2 Observational data and quality control**

A PARSIVEL is a disdrometer that uses a laser beam with a wavelength of 780 nm to obtain a particle's equivolume diameter
130     ($D$, mm) and fall velocity ($V_f$, m s$^{-1}$) based on changes in the laser beam signals. The measurable range of $D$ ($V_f$) is from 0.3 mm (0.1 m s$^{-1}$) to 30 mm (20 m s$^{-1}$). The overall error of $D$ is within 5% and $V_f$ has errors ranging from 10 % to 25% as $D$ changes (Löffler-Mang and Joss, 2000). We suggest how to deal with these measurement errors in Section 3.1. Version 2 PARSIVELs and level 1 data are used in the present study. Level 1 data are format-converted with no processing and provide particle counts for individual diameter and velocity channels (a 32 by 32 array) every 1 min. Because the observed PARSIVEL
135     data contain outliers that may be the result of various forms of error, such as calibration errors and "margin fallers" (Yuter et al., 2006), we eliminate any of the level 1 data that meet one or both of the following two criteria: i) $D < 1$ mm and ii) $V_f > 1.4V_a$. $V_a$ is the empirical relationship between $D$ and $V_f$ established by Atlas et al. (1973).

A modem-type rawinsonde (M10) is used for the ICE-POP 2018 campaign (In et al., 2018). The observation variables recorded by the M10 rawinsonde are pressure ($P$, hPa), $T$ (°C), RH (%), wind speed (WS, m s$^{-1}$) and wind direction (WD, °)
140     at 1 s intervals. Additionally, $T_w$ is calculated using the two-parameter relationship for $T$ and RH suggested by Stull (2011). Although rawinsonde data is useful as a reference of atmospheric vertical structure, the absolute accuracy of $T$ and RH of the M10 rawinsonde sensor are 0.3°C and 3%, respectively (In et al. 2018). The impact of these measurement errors can be significant near 0 °C, where phase changes of precipitation particles occur.

The MRRs are modulated continuous wave (FMCW) radar instruments using a solid-state transmitter with a frequency of
145     24 GHz (Maahn and Kollias, 2012). In this study, the range resolution of the MRRs is set to 150 m. This resolution is enough to identify the ML because the average ML depth based on dual-polarization radar measurements from the Korean peninsula during winter is about 670 m (Allabakash et al. 2019). Raw data from MRR supplies vertical profiles of radar reflectivity ($Z$, dBZ) and Doppler velocity ($V_r$, m s$^{-1}$) in precipitation. $Z$ and $V_r$ can be contaminated by noise including non-meteorological echoes. Also, if $V_r$ exceeds the Nyquist velocity boundaries (-6 m s$^{-1}$ ~ 6 m s$^{-1}$) of the MRR, aliasing of $V_r$ will occur (Maahn
150     and Kollias, 2012). In general, large raindrops in heavy rainfall events cause the aliased data. Therefore, raw data from the MRRs are quality-controlled using de-aliasing and the noise removal algorithm suggested by Maahn and Kollias (2012). The processed MRR data are used to provide additional context for important cases in the present study.

**2. Lines 156-162: Was there only one sounding available for each precipitation event? Should the earlier soundings be used as environmental profiles to diagnose precipitation types?**

➔ Yes. One event is only one sounding data. If precipitation is identified by PARSIVEL when sounding launches at specific time and site, the event includes matched precipitation case(The explanation about selection of 'matched precipitation case' is added at main contents). 'Winter precipitation type' is mainly decided by low-level atmospheric condition (melting layer, freezing layer, inversion layer, saturation layer, and so on). Low-level atmospheric condition is very changeable. Therefore, when precipitation occurs, it is appropriate to use sounding data from the nearest time or the current time for diagnosis. Sounding data of current time in this study was used.

(red color letter between 174~175 lines of revision file)

We obtain a total of 131 matched precipitation cases to validate the five diagnostic methods during the ICE-POP period (1 November 2017–30 April 2018). If precipitation is identified when a sounding launches at a specific time and site, the event
175 includes a matched precipitation case. Cases are identified that feature measurable precipitation at each of the five sounding sites. We identify precipitation cases at each site that satisfy two conditions: i) $N_{RA}+N_{SN} \geq 15$ within 5 min of the sounding start time, and ii) $-4\ °C < T_0 < 6\ °C$ and $RH_0 > 40\ \%$ at the sounding start time. Here, $T_0$ and $RH_0$ are the data recorded 1 s after the start of the sounding, and they accurately represent the surface $T$ and $RH$ measured by the rawinsonde. Based on this hydrometeor-type classification scheme, the dominant WPT of the matched precipitation cases is determined using the newly
180 developed algorithm with the quality-controlled 5 min PARSIVEL data (Fig. 2b).

**3. Lines 221-222: How to determine critical values for different sites? Please clarify.**

➔ Thank your advice and I reflect it. The same critical value was used for sites with similar terrain characteristics. I add supplementary explanation at main contents.

(red color letter between 241~243 lines of revision file)

240 gpm, respectively. The WPTs at GWU, BKC, and SCW are diagnosed using the former critical values, while DGW and MOO are diagnosed using the latter. The reason for using different critical values is that GWU, BKC, and SCW are located near the East sea at a low altitude and east of the Taebaek mountains whereas DGW and MOO are located within the Taebaek mountains and have a relatively higher altitude (Fig. 1a).

**4. Lines 293-294: Please justify "initialized as unrimed low-density snow aggregates".**

➔ Thank advice, I should add explanation. In this study, Number of SN events is 91, graupel-like events is only 13 events and not-graupel events is 78. We can divide this by using 'F val'. You can see results at "Surface precipitation type based on new decision algorithm about 131 precipitation events in 5 ICE-POP 2018 sites" asset https://doi.org/10.5281/zenodo.13561536

In the fig (PARSIVEL V-D scatterplot of each event), you can see 'F val'.

[Figure]

We calculated 'F val' following Lee et al. (2015), 'F val' quantifies difference between data and empirical relationship. So, F value is smallest, it is dominant hydrometeor (marks red color).

I added explanation at main contents.

(red color letter between 315~317 lines of revision file)

315    The SBM parameters used in this study are presented in Table 2. The particles are separated into 20 size bins and initialized as 'unrimed low-density snow aggregates' because there are only 13 graupel-like events among the 91 SN events following the hydrometeor classification method suggested by Lee et al. (2015). The size bins are delineated such that the equivolume diameters of fully melted particles of equal mass in each bin are 0.1 mm apart. The largest size bin used in this study, with a fully melted equivolume diameter $D_{mw,max}$ of 1.95 mm, is about 2 times the mean value of the mass-weighted mean diameter

Lee, J. E., Jung, S. H., Park, H. M., Kwon, S., Lin, P. L., and Lee, G.: Classification of precipitation types using fall velocity-diameter relationships from 2D-video distrometer measurements, Adv. Atmos. Sci., 32, 1277-1290., https://doi.org/10.1007/s00376-015-4234-4, 2015.

**5. Lines 298-303: The authors argued that "the assumption of mass conservation" may be valid. However, how about PSDs? Given the same mass, PSDs at the surface and in the upper atmosphere could differ significantly. Please justify it.**

➔ Yes, actually, PSD during falling is very changeable and continuously evolved by many microphysical processes (aggregation, riming, and so on). And, aircraft microphysics data is very useful for initial PSD. However, aircraft microphysics data is very lack during ICE-POP period. Also, current SBM scheme not include aggregation and rimming process. I add sentence about this situation at main content.

(red color letter between 327~328 lines of revision file)

325    evaporation/sublimation) for simplicity and instead only consider melting/refreezing. The assumption of mass conservation should generally be valid for this study because almost all of the precipitation cases are nearly saturated (RH > 80 %) below 5 km AGL (Fig. 5b). The initial PSD is fixed for all events because of the lack of aircraft microphysical observation data and the exclusion of aggregation/riming process in the microphysics scheme in the current SBM.

**6. Figures 10-12: Which SBM method, original or optimized one is shown in these figures?**

➔ Thank your check. I specifies it at caption in Fig. 10.

And, from request of editor (specify model 'version'), model version is specified in introduction section. SBM have 3 versions:

1) Reeves et al (2016): origin version

2) Carlin et al (2019): 1DSBM-19 (upgrade of Reeves et al. 2016)

3) This study: 1DSBM-19M (modified version of Carlin et al. 2019)

Because 'original SBM' in from 'data' section to 'summary and future work' section means 1DSBM-19M, I change expression 'original SBM' to 'current SBM'.

(red color letter between 478 line of revision file)

475                                      $T_w$ [°C]

**Figure 10.** $T_w$ profiles for observed SN cases occurring at (a) mountain sites and (b) coastal sites. The blue, red, and green lines indicate diagnosed RA, SN, and RASN cases, respectively, for the $H_{850}$ thickness, shifted Matsuo scheme on $RH_0$ - $T_0$ nomogram, wet-bulb temperature $T_{w0}$, $T_{w0}$-$\Gamma_{low}$ nomogram, and the current SBM methods. Bold lines indicate misdiagnosed cases.

**7. Line 500: Why were not all examples within each group included, especially given the limited number of examples? A justification for this selection would be helpful.**

➜ A representative example from each group is only shown because each group have similar atmospheric environmental characteristics (I add this sentence at main contents). Testing was conducted at all eight misdiagnosed cases. Simulation results (correct/not correct of precipitation type) of eight misdiagnosed cases was mentioned at 584~585 lines (revision file):

 'Among the eight misdiagnosed cases in the SBM, four are correctly diagnosed by the optimized SBM. If a more accurate 585 cloud top is also considered, two more cases are correctly diagnosed by the optimized SBM.).'

(red color letter between 525~526 lines of revision file)

525    Only a representative example from each group is shown because each group has similar atmospheric environmental characteristics. We also compare the simulation results between the current and optimized SBMs. Figure 13 presents the environmental profiles from the rawinsondes and $V_f$–$D$ scatterplots from the corresponding PARSIVEL taken at around 1200

**8. Lines 546-547: Should the authors also consider updating the Vt-D relationship for ice particles?**

➜ Although event 3 at Fig.14 was not refreezing event (based on Fig. 15), I also think evaluation of SBM microphysics scheme about ice pellet event will be required. I add this at 'summary and future work' section.

(red color letter between 618~619 lines of revision file)

The potential of the SBM for diagnosing the WPT was thus confirmed in the present study. The performance of the current SBM was superior to existing optimized methods (the $H_{850}$ and $RH_0$-$T_0$ nomogram methods) and the skill scores were improved further via regional optimization of the SBM's microphysics scheme. Furthermore, there is a need to verify the microphysics

scheme in the SBM in more detail, such as for IP events and so on. We will focus on the development of a combined SBM
620    with other reanalysis field data for the acquisition of three-dimensional WPT information.

---

## Community Comment (CC3)

**Response to Reviewer**

**19 December 2024**

**Overview:**

**This paper is an evaluation of precipitation type diagnosis algorithms in a region of complex terrain and coastal influences in South Korea.   In general, the paper is well written, and the results are clearly explained.   I think that the paper is ready for publication after the authors address some minor issues.**

Dear Referee,

We greatly appreciate your positive feedback and the time and effort you devoted to reviewing our manuscript and dataset. We have carefully reviewed your comments and actively reflected them. Thank you again for your help in improving our manuscript.

Best regards,

Wonbae Bang (on behalf of the author team)

**Specific Comments:**

**1. Line 34: Does vaporization = evaporation?  I would recommend using evaporation here (as already used elsewhere in the paper), as it is more commonly used in meteorology and will be more familiar to readers.**

➔ thank you for confirming. I modify it.

(skyblue color letter at 35 line of revision file)

> There is a complex variety of winter precipitation types (WPTs) such as rain (RA), snow (SN), rain and snow (RASN), ice pellets (IPs), freezing rain (FZRA), and a mixture of ice pellets and freezing rain (IPFZRA). Various thermodynamical and microphysical processes can determine surface WPTs in nature. Some microphysical processes, such as melting, freezing,
> 35 evaporation, and sublimation, change the phase and/or mass of precipitation particles and are diabatic thermodynamic

**2. Lines 41-77: I think it would be worthwhile to mention precipitation type diagnosis algorithms that work in conjunction with microphysical parameterizations within numerical weather prediction models.  For example, the algorithm described in this paper:**

**https://doi.org/10.1175/WAF-D-15-0136.1**

**If you briefly described those algorithms, you could distinguish them from the types of algorithms you are evaluating herein (which are based purely on observations).**

➔ Thank you for recommending this good paper. I describe algorithm of Benjamin et al. (2016) at Introduction.

(skyblue color letter between 59~63 lines of revision file)

> 55 account situations where the melting of ice particles begins while they are falling, which is especially important for conditions that include low-level temperature inversions. However, because this scheme was developed using global data without regional and/or synoptic weather dependence, it is only valid when used in a globally averaged manner. The validity for the regions of this study has not been investigated in Sims and Liu (2015). In addition to those described here, many other WPT diagnostic methods based on the environment or numerical model data have been proposed (e.g., Ramer, 1993; Baldwin et al., 1994;
> 60 Bourgouin, 2000; Schuur et al., 2012; Benjamin et al., 2016). As an example, Benjamin et al. (2016) suggested diagnostic logic for WPT using output of the Rapid Refresh (RAP) and High-Resolution Rapid Refresh (HRRR) models such as 2-m $T$, total precipitation, precipitation except graupel, snow-only precipitation, snow fraction, precipitation rate, and so on. The diagnostic logic classifies four WPTs (RA, SN, FZRA, IP) based on a decision tree method.

**3. Lines 309-321: Can you explicitly describe h. Is it the overall hit rate? Whereas h' is averaged across three p types? So if one p-type does particularly badly, but only has a few cases, h' will be much lower than h. Is that right? I think the distinction between h and h' could be more clearly described, which would help the reader interpret results.**

➔ Yes, your understanding is correct. I also feels description of h, h' in manuscript is insufficient, I modifies more specifically the sentence.

(skyblue color letter between 343~345 lines of revision file)

where $O$ is the number of observed cases, and $E$ is the number of correctly diagnosed cases from among the observed cases for each method. We calculate the $h$, $h_{SN}$, $h_{RASN}$, $h_{RA}$, and $h'$ for each of the diagnosis methods. Here, $h$ without a subscript is the overall hit rate. $h$ with a subscript (SN, RASN, RA) represents the accuracy for each WPT type, while $h'$ is the average

345    accuracy across all three WPTs. The skill scores are also compared between the mountain sites (DGW and MOO) and coastal sites (GWU, SCW, and BKC), and the effect of vertical $T_w$ profiles on the accuracy of each diagnosis method is investigated to assess the strengths and weaknesses of each diagnosis method.

**4. Lines 525-527: It seems like collision-coalescence is an important factor to include in SBM. Can you provide some more detail on its effects and reasons for exclusion?**

➔ Yes, Collision-coalescence (C-C) process is also an important factor because C-C process change drop size distribution. However, considering collision-coalescence in SBM is area not yet developed. I add a sentence about importance of C-C process and a sentence about including C-C process in the development direction of SBM.

(skyblue color letter between 551~554 lines of revision file)

optimized SBM simulation significantly increases the amount of melting. However, the maximum diameter with complete melting ($\sim 1.35$ mm) in the simulation is still slightly smaller than the observed $D_{max}$ ($\sim 1.62$ mm). This difference could be

550    the result of three sources of error: northward advection of the rawinsonde due to low-level southerly winds, hardware

calibration issues for the GWU PARSIVEL, and/or the growth of raindrops via a collision–coalescence process. Collision-coalescence is also an important factor for the classification of WPT because the process increases the average raindrop size and decreases the number concentration of small drops. However, this process is not currently included in the SBM owing to algorithm efficiency demands but it is a major area for future improvement.

555

---

## Author Response (AR1)

**Response to Reviewer 1**

**31 January 2025**

**Overview:**

**The authors evaluated five diagnosis schemes of identifying winter precipitation types using data from the ICE-POP 2018 field experiment. They found that the scheme using one-dimensional spectral bin model (SBM) with the climatological snow density-diameter relationship for the Pyeongchang region demonstrates superior performance. The manuscript is well written, clear, and easy to follow. I have only some minor comments regarding clarification or justification for consideration.**

Dear Referee,

We greatly appreciate your positive feedback and the time and effort you devoted to reviewing our manuscript and dataset. We have carefully reviewed your comments and actively reflected them. Thank you again for your help in improving our manuscript.

Best regards,

Wonbae Bang (on behalf of the author team)

**Specific Comments:**

**1. Section 2.2: How large uncertainties of these observations? The authors should discuss them to enhance the manuscript's robustness.**

➔ Thank your advice and I reflect it. I add explanation about measurement errors of PARSIVEL and sounding. Also, specification of MRR is more detailed at Section 2.2.

(red color letter between 129~150 lines of revision file)

**2. Lines 156-162: Was there only one sounding available for each precipitation event? Should the earlier soundings be used as environmental profiles to diagnose precipitation types?**

➔ Yes. One event is only one sounding data. If precipitation is identified by PARSIVEL when sounding launches at specific time and site, the event includes matched precipitation case(The explanation about selection of 'matched precipitation case' is added at main contents). 'Winter precipitation type' is mainly decided by low-level atmospheric condition (melting layer, freezing layer, inversion layer, saturation layer, and so on). Low-level atmospheric condition is very changeable. Therefore, when precipitation occurs, it is appropriate to use sounding data from the nearest time or the current time for diagnosis. Sounding data of current time in this study was used.

(red color letter between 174~175 lines of revision file)

**3. Lines 221-222: How to determine critical values for different sites? Please clarify.**

➔ Thank your advice and I reflect it. The same critical value was used for sites with similar terrain characteristics. I add supplementary explanation at main contents.

(red color letter between 238~240 lines of revision file)

**4. Lines 293-294: Please justify "initialized as unrimed low-density snow aggregates".**

➔ Thank advice, I should add explanation. In this study, Number of SN events is 91, graupel-like events is only 13 events and not-graupel events is 78. We can divide this by using 'F val'. You can see results at "Surface precipitation type based on new decision algorithm about 131 precipitation events in 5 ICE-POP 2018 sites" asset https://doi.org/10.5281/zenodo.13561536

In the fig (PARSIVEL V-D scatterplot of each event), you can see 'F val'.

[Figure]

We calculated 'F val' following Lee et al. (2015), 'F val' quantifies difference between data and empirical relationship. So, F value is smallest, it is dominant hydrometeor (marks red color).

I added explanation at main contents.

(red color letter between 309~311 lines of revision file)

Lee, J. E., Jung, S. H., Park, H. M., Kwon, S., Lin, P. L., and Lee, G.: Classification of precipitation types using fall velocity-diameter relationships from 2D-video distrometer measurements, Adv. Atmos. Sci., 32, 1277-1290., https://doi.org/10.1007/s00376-015-4234-4, 2015.

**5. Lines 298-303: The authors argued that "the assumption of mass conservation" may be valid. However, how about PSDs? Given the same mass, PSDs at the surface and in the upper atmosphere could differ significantly. Please justify it.**

➔ Yes, actually, PSD during falling is very changeable and continuously evolved by many microphysical processes (aggregation, riming, and so on). And, aircraft microphysics data is very useful for initial PSD. However, aircraft microphysics data is very lack during ICE-POP period. Also, current SBM scheme not include aggregation and rimming process. I add sentence about this situation at main content.

(red color letter between 321~322 lines of revision file)

**6. Figures 10-12: Which SBM method, original or optimized one is shown in these figures?**

➔ Thank your check. I specifies it at caption in Fig. 10.

And, from request of editor (specify model 'version'), model version is specified in introduction section. SBM have 3 versions:

1) Reeves et al (2016): origin version

2) Carlin et al (2019): 1DSBM-19 (upgrade of Reeves et al. 2016)

3) This study: 1DSBM-19M (modified version of Carlin et al. 2019)

Because 'original SBM' in from 'data' section to 'summary and future work' section means 1DSBM-19M, I change expression 'original SBM' to 'current SBM'.

(red color letter between 487 line of revision file)

**7. Line 500: Why were not all examples within each group included, especially given the limited number of examples? A justification for this selection would be helpful.**

➔ A representative example from each group is only shown because each group have similar atmospheric environmental characteristics (I add this sentence at main contents). Testing was conducted at all eight misdiagnosed cases. Simulation results (correct/not correct of precipitation type) of eight misdiagnosed cases was mentioned at 584~585 lines (revision file):

 'Among the eight misdiagnosed cases in the SBM, four are correctly diagnosed by the optimized SBM. If a more accurate 585 cloud top is also considered, two more cases are correctly diagnosed by the optimized SBM.).'

(red color letter between 536~537 lines of revision file)

**8. Lines 546-547: Should the authors also consider updating the Vt-D relationship for ice particles?**

➔ Although event 3 at Fig.14 was not refreezing event (based on Fig. 15), I also think evaluation of SBM microphysics scheme about ice pellet event will be required. I add this at 'summary and future work' section.

(red color letter between 629~630 lines of revision file)

**Response to Reviewer 2**

**31 January 2025**

**Overview:**

**This paper is an evaluation of precipitation type diagnosis algorithms in a region of complex terrain and coastal influences in South Korea.  In general, the paper is well written, and the results are clearly explained.  I think that the paper is ready for publication after the authors address some minor issues.**

Dear Referee,

We greatly appreciate your positive feedback and the time and effort you devoted to reviewing our manuscript and dataset. We have carefully reviewed your comments and actively reflected them. Thank you again for your help in improving our manuscript.

Best regards,

Wonbae Bang (on behalf of the author team)

**Specific Comments:**

**1. Line 34: Does vaporization = evaporation?   I would recommend using evaporation here (as already used elsewhere in the paper), as it is more commonly used in meteorology and will be more familiar to readers.**

➔ thank you for confirming. I modify it.

(skyblue color letter at 35 line of revision file)

**2. Lines 41-77: I think it would be worthwhile to mention precipitation type diagnosis algorithms that work in conjunction with microphysical parameterizations within numerical weather prediction models.   For example, the algorithm described in this paper:**

**https://doi.org/10.1175/WAF-D-15-0136.1**

**If you briefly described those algorithms, you could distinguish them from the types of algorithms you are evaluating herein (which are based purely on observations).**

➔ Thank you for recommending this good paper. I describe algorithm of Benjamin et al. (2016) at Introduction.

(skyblue color letter between 59~63 lines of revision file)

**3. Lines 309-321: Can you explicitly describe h. Is it the overall hit rate? Whereas h′ is averaged across three p types? So if one p-type does particularly badly, but only has a few cases, h′ will be much lower than h. Is that right? I think the distinction between h and h′ could be more clearly described, which would help the reader interpret results.**

➔ Yes, your understanding is correct. I also feels description of h, h′ in manuscript is insufficient, I modifies more specifically the sentence.

(skyblue color letter between 335~337 lines of revision file)

**4. Lines 525-527: It seems like collision-coalescence is an important factor to include in SBM. Can you provide some more detail on its effects and reasons for exclusion?**

➔ Yes, Collision-coalescence (C-C) process is also an important factor because C-C process change drop size distribution. However, considering collision-coalescence in SBM is area not yet developed. I add a sentence about importance of C-C process and a sentence about including C-C process in the development direction of SBM.

(skyblue color letter between 563~565 lines of revision file)

**Response to Reviewer 3**

31 January 2025

**Overview:**

**The manuscript compared the performance of five different schemes for identifying winter precipitation types (limited to rain, snow, and mixed rain-snow) using data from the ICE-POP 2018 field experiments. The study demonstrated that an enhanced spectral bin model (SBM) provided relatively better results. While the manuscript contains valuable analyses and experimental results, the current presentation and English writing do not meet the publication quality. A major revision is recommended. Thorough proofreading will improve the manuscript greatly. Many places require further editing and some editing examples have been provided.**

Dear Referee,

We sincerely appreciate the time and effort you devoted to reading our study and providing thoughtful, detailed feedback. Your insightful comments and constructive suggestions have significantly contributed to enhancing the clarity, depth, and overall quality of my work. Thank you for sharing your expertise and for your valuable guidance throughout this process. Your input has been invaluable in shaping our research, and we are grateful for your support.

Best regards,

Wonbae Bang (on behalf of the author team)

**Major Comments:**

**1. The "hit rate" alone does not reveal the full picture. Suggest adding the False Alarm Rate (FAR) to get a better picture of the performance of different schemes**

We add additional skill scores considering 'false alarms': CSI (critical success index) and FAR (false alarm rate). CSI and FAR are calculated about 9 categories (3 precipitation types for 3 regions — all/mountain/coastal) and Table 3, 4 are included in section 4.1 Overall accuracy of the diagnosed precipitation types. Also, the calculation method of CSI and FAR is included in section 3.3 Evaluation method. Section 5. Summary and future work section is also modified.

(orange color text between lines 337~343, 419~429, 607, and 624~625 in the revision file)

**2. There are too many uncommon acronyms in the manuscript. Suggest reverting some of them back to their original forms as much as possible (acronyms in equations may be kept) to help readers understand easily.**

**WL -> warm layers, IL -> warm layer, CL -> cold layer, R-> rainfall rate, SR -> snowfall rate, etc**

**Also, it will be good to add a "List of Acronyms"**

We reverted WL, IL, and CL, but maintained our use of R and SR because they are widely used expressions. We included a 'List of Acronyms' between 'code and data availability' and 'author contribution'.

(orange text on line 645 in the revision file)

**3. The hydrometeor falling velocity is usually called the "terminal velocity" and referred to as "Vt" instead of "Vf".**

We reverted the expression following your suggestion.

**4. Lines92-93: "Tw-Γlow nomogram (Sims and Liu, 2015) methods" -> Need to be consistent in the manuscript on how to refer to this method, either "the Tw-Γlow method" or "the Sims and Liu method". Avoid using different names in different locations (such as lines 355, 566 etc) for the same method. Revise similarly for other methods throughout the manuscript.**

. We agree that "consistence for naming method should be required", we modified "relating words" in this way: "variable"+ method.

**Line 24: "The results show that the SBM has the highest overall skill score for winter precipitation": SBM did not outperform all other 4 schemes in all situations as demonstrated in the manuscript. Please revise accordingly.**

We revise the contents more exactly.

(orange text between lines 24~26 in the revision file)

**Line 29: "which uses climatological relationships for Colorado region" -> "which uses a snow density-diameter relationship for the Colorado region".**

We modified the text following your suggestion.

(orange text between lines 29~30 in the revision file)

**Line 73: "The addition of sublimation and evaporation is predicated on the idea that these processes may..." –> "The addition of sublimation and evaporation is because these processes may..."**

We modified the text following your suggestion.

**Lines 87-88: "the intensive observation data density..." –> "The high-density observational data provided by the ICE-POP network enables a comprehensive evaluation and refinement of previously proposed WPT diagnostic methods."**

We modified the text following your suggestion.

**Lines 142-143: "a low-speed mode ... of graupel or hail":  Why hail has a low fall velocity (terminal velocity)?**

Following Nagumo and Fujiyoshi (2015), the liquid water fraction of an ice pellet relates to the terminal fall velocity mode. If the surface of an ice pellet is frozen but the inside of the ice pellet remains water, its terminal fall velocity is similar to that of a raindrop (i.e., high-speed mode). If most of the ice pellet is frozen, the ice pellet will havea similar fall speed to small hail or graupel (i.e., low-speed mode).

Reference

Nagumo, N., and Y. Fujiyoshi. Microphysical properties of slow-falling and fast-falling ice pellets formed by freezing associated with evaporative cooling, Mon. Wea. Rev., 143(11), 4376-4392, https://doi.org/10.1175/MWR-D-15-0054.1, 2015.

**Fig. 2:   "Fsite" is not defined in Fig. 2b**

We added an explanation about Fsite.

(orange text between lines 189~190 in the revision file)

**Line 172: How is the "normalized frequency" computed?**

We added an explanation about the calculation of normalized frequency.

(orange text between lines 184~185 in the revision file)

**Line 177: why is a threshold value of "0.05" critical? How do we get this threshold?**

Although we select only pure rain events, observation/measurement errors may exist and sometimes can produce outlier values. A threshold of 0.05 was chosen to exclude these outliers.

**Lines 249-250: "For example, ...": (1) Based on Fig.4d, Zero Tw and a lapse rate of 6 °C/km correspond to about 80% probability. Where did the "probability of 0.86" come from? "0.45" is not accurate either, (2) Suggest adding a zero Tw line to help users examine Fig4d.**

We added grid lines in Fig.4d, and changed examples to easily readable values.

(orange text between lines 266~268 in the revision file)

**Lines 259-261: This sentence is redundant or it should appear where the Tw0-Гlow method was first introduced.**

We simplified the sentence and complemented the first introduction of the Tw0-Гlow method by including an additional explanation.

(orange text on lines 52~53, 275 in the revision file)

**Line 286: How were the rainfall rate and the snowfall rate data obtained?**

The SBM simulates PSD evolution from the cloud top (initial PSD is following Table 2) to ground level. Rainfall rate and snowfall rate are calculated from the simulated ground PSD. We complement the sentence by including an additional explanation.

(orange text between lines 300~301 in the revision file)

**Lines 408-413: "For RASN, ...": The last sentence should be moved to the location before "For RASN..." and then revise the remain part accordingly.**

We changed the sequence of the sentences.

(orange text between lines 445~447 in the revision file)

**Line 314: What's the purpose of using h'? It looks like h' is not needed since the hit rates were calculated individually for SN, RASN and RA and plotted in figures (such as Fig. 6) which are enough for the discussions.**

Because of the different number of events of each type (SN: 90, RASN: 15, RA: 26 in Table 1), the overall h value can be distorted. h' is less influenced by the different number of events of each type and serves as a complement to the h score. For example, consider the case where a specific method has predictions of SN and RA are perfect but RASN has no hits. In that case, h is very high (88.5%) whereas h' is 66.7%. Introducing h' can make a balanced evaluation.

**Line 340: "We evaluate the accuracy of the H850 method": "Accuracy" has a special meaning and formula in statistics. Suggest changing the "accuracy" to "performance" or similar words throughout the manuscript.**

We modified the sentences to use 'evaluate' and 'performance' together.

**Lines 346-351: What is the conclusion from this discussion (about different thresholds for the Tw0-Γlow method)? If there are no conclusions from this discussion, it may be removed.**

We tested various critical values (10% 20% 30%, 0.1 0.2 0.3) for diagnosing winter precipitation type by using the Tw/Tw - Γlow methods. Therefore, we displayed performance using the 20% or 30% (0.2 or 0.3) critical values together for readers. Tw has the best performance at 10%, and Tw – Γlow with 0.1 has very stable results (h and h' are very similar values) and the best h'. Therefore, we used this to decide the critical values (10% for Tw, 0.1 for Tw – Γlow).

**Lines 364-365: "The Tw0 method exhibits a relatively large difference between h (86.3 %) and $h'$ (68.4 %), with the inclusion of Tw0 improving the diagnosis of SN (Figs. 6c and 6d).": (1) it discussed two conclusions in one sentence and does not read well; (2) this sentence can be removed without any evident impact.**

Because the accuracy of SN for methods that include $T_{w0}$ ($T_{w0}$ method, $T_{w0}$-$\Gamma_{low}$ method) is much higher than those that do not include $T_{w0}$ ($H_{850}$ and $RH_0$-$T_0$ methods). We supplement the sentence.

(orange text between lines 377~378 in the revision file)

**Lines 364-367: The low performance in diagnosing RASN in Fig. 6c (the Tw0 method) should be highlighted here and h' is not needed.**

We deleted 'reducing $h'$' and include the skill score for RASN in the sentence.

(orange text between lines 378~379 in the revision file)

**Lines 367-368: "This is supported by...(Sims and Liu, 2015)." This logic here is not clear. Consider revising.**

The following sentence is the reason (The 90 % conditional probability of SN for land areas varies from $T_{w0} = -0.1$ °C at $\Gamma_{low} = 11$ °C km$^{-1}$ to $T_{w0} = -4.1$ °C at $\Gamma_{low} = -5$ °C km$^{-1}$). Therefore, the critical value ($T_{w0} = 0.5$ °C ) is warmer than the statistics indicated. We have improved the logical flow of the relevant sentences.

(orange text between lines 379~381 in the revision file)

**Fig. 6: "0(0.0%)" labels can be removed from the figure.**

We modified this following your suggestion.

**Lines 391-395: "The dependence of the skill scores on the terrain for all five methods is also explored...: The paragraph before this one already discussed the impact of terrain on the skill scores. Consider revising to make the logic smoother or merging this paragraph to the previous one.**

We modified this following your letter suggestion (merging this paragraph).

**Lines 395-435: These paragraphs did not discuss the "dependence on wet-bulb temperature profiles" and hence should be placed before section 4.2. Either put them in section 4.1 or add a new section.**

In this section, the data distribution analysis of Fig. 9 is very important for interpretation of Fig.10~12 because characteristics of the Tw profile strongly influence values of H850, RH0-T0, Tw0, Tw0-$\Gamma_{low}$.

**Line 407: "In contrast": I don't think there is an evident "contrast" here. It is safe to remove it.**

We modified the text following your suggestion.

**Lines 417-420: Two sentences here, and the second one repeats the first one.**

We modified the text following your suggestion.

**Line 425: "The Γlow of RA varies widely, though it tends toward negative values": Fig. 9d showed that most Γlow values are positive. So why it is stated "tends toward negative values"?**

This part was a mistake. We have modified negative -> positive.

**Fig. 10: remove the "RA" legend as there are no RA cases in this figure.**

We modified the figure following your suggestion.

**Lines 428 and 474: What is "complex atmospheric profiles" or "complex atmospheric vertical structure"? Need clarification on this.**

"Complex atmospheric vertical structure" refers to profiles with a melting layer aloft and a near-surface refreezing layer. We have expanded upon the sentence.

(orange text between lines 462~463 in the revision file)

**Table 3: Need a separating line between Group 1 and Group 2**

We modified the table following your suggestion.

**Fig. 13a,b,c: No discussion on these three figures in the manuscript. If not needed, they can be removed.**

Lines 504~508 already include a description of Fig. 13a,b,c. (submitted file)

**Fig. 14b, f: NO discussion on these two figures. If not needed, they can be removed.**

Lines 521~522 already include a description of Fig. 14b,f. (submitted file)

**Fig 14e,h: No description of these two figures in the figure caption.**

Lines 520~521 already include a description of Fig. 14e, while lines 547~548 include description of Fig. 14h. (submitted file)

**Line 540 Why define "Doppler velocity" as -Vr instead of Vr?**

Our intention is for the Doppler velocity to be represented with the same sign with Vt.

**Line 542: Figs. 15c and 15d  -> Figs. 15a and b**

We modified the text following your suggestion.

**Lines 582-583: Do you mean "These results suggested that SBM simulations tend to produce less melting compared to the observed precipitation"?**

Yes, that's correct. We modified the text following your suggestion.

**Lines 590-591: "The performance of the original SBM was superior to some existing optimized methods (the H850 and RH0-T0 nomogram methods)"  ->  add "some" before "existing" since SBM did not outperform all methods in all situations**

We modified the text following your suggestion.

**Lines: 592-593: (1) "should" -> "will";   (2) what does "other reanalysis filed data" refer to? (3) how would the 3D WPT algorithm differ from the 2D WPT algorithm?**

We correct should->will, and complement the sentence: other 3-dimensional reanalysis field data (Local Data Assimilation and Prediction System), and include additional sentence about utilization of 3D WPT.

(orange text between lines 630~633 in the revision file)

**Edits:**

**Line2 145-146: "very strong inversion strength"   -> "very strong inversion"**

We modified the text following your suggestion.

**Line 269: "the next largest size" -> "the next larger size"**

We modified the text following your suggestion.

**Line 340: "an RH0-T0 nomogram" -> "the RH0-T0 method"**

We modified the text following your suggestion.

**Line 341: "a Tw0-Гlow nomogram" -> "the Tw0-Гlow method", revise similarly throughout the manuscript**

We modified the text following your suggestion.

**Line 343: "The lowest h is achieved by the RH0-T0 nomogram" -> "The lowest h is from the RH0-T0 method"**

We modified the text following your suggestion.

**Line 400: "radiational cooling" -> "radiative cooling"**

We modified the text following your suggestion.

**Line 420: "The Tw0 for cases" -> "The Tw0 for RASN cases"**

We modified the text following your suggestion.

**Line 428: "A cold RASN case" -> "a RASN case"**

We modified the text following your suggestion.

---

## Author Response (AR2)

**Response to Reviewers**

**17 Mar 2025**

**Dear Referees,**

**The comments from Reviewer 2 have been reflected in blue in the revised manuscript, while the comments from Reviewer 3 have been reflected in red. Once again, thank you for your valuable feedback and for helping us improve the quality of our manuscript.**

**Best regards,**

**Wonbae Bang (on behalf of the author team)**

**Report #2**

**I would say "decision tree algorithm" instead of "decision algorithm" in the abstract and in the text (line 568 and caption for Fig. 2). It is a clearer phrase than "decision algorithm".**

We modified the text following your suggestion.

➔ (line 24, 194, 603, 639 in revision file)

**Line 523: there is hardware errors > there are hardware errors**

We modified the text following your suggestion.

➔ (line 523 in revision file)

**Line 632: Full of help > helpful**

We modified the text following your suggestion.

➔ (line 632 in revision file)

**Report #3**

**Line 160: change "graupel or hail" to "graupel or small hail"**

We modified the text following your suggestion.

➔ (line 160 in revision file)

---

## Author Response (AR3)

**Response to editors**

**25 Mar 2025**

Dear Daria Karpachova,

First of all, thank you for carefully reviewing our manuscript. We have improved our manuscript according to your advice. For Figures 9 to 12, the line thickness of the red-toned and green-toned data has been differentiated to ensure that colorblind individuals can distinguish the data more easily. The names and order of the manuscript chapters have been edited to comply with the journal's guidelines. "Appendix" has been revised to "Appendix A," and its position has been moved to appear before the "Code and Data Availability" section.

Best regards,

Wonbae Bang (on behalf of the author team)